# Cell-cycle quiescence maintains *Caenorhabditis elegans* germline stem cells independent of GLP-1/Notch

**Hannah S Seidel[1,2]\*, Judith Kimble[1,3]\***

[1]Department of Biochemistry, University of Wisconsin-Madison, Madison, United States; [2]The Ellison Medical Foundation Fellow of the Life Sciences Research Foundation, The Lawrence Ellison Foundation, Mount Airy, United States; [3]Howard Hughes Medical Institute, University of Wisconsin-Madison, Madison, United States

**Abstract** Many types of adult stem cells exist in a state of cell-cycle quiescence, yet it has remained unclear whether quiescence plays a role in maintaining the stem cell fate. Here we establish the adult germline of *Caenorhabditis elegans* as a model for facultative stem cell quiescence. We find that mitotically dividing germ cells—including germline stem cells—become quiescent in the absence of food. This quiescence is characterized by a slowing of S phase, a block to M-phase entry, and the ability to re-enter M phase rapidly in response to re-feeding. Further, we demonstrate that cell-cycle quiescence alters the genetic requirements for stem cell maintenance: The signaling pathway required for stem cell maintenance under fed conditions—GLP-1/Notch signaling—becomes dispensable under conditions of quiescence. Thus, cell-cycle quiescence can itself maintain stem cells, independent of the signaling pathway otherwise essential for such maintenance.

\*For correspondence: hsseidel@ wisc.edu (HSS); jekimble@wisc.edu (JK)

**Competing interests:** The authors declare that no competing interests exist.

## Introduction

Stem cells in adult tissues were once thought to exist primarily in a state of cell-cycle quiescence. Such quiescence was viewed as an inherent property of the stem cell fate and thus essential for a tissue's long-term self-renewal (*Hall and Watt, 1989*; *Potten and Loeffler, 1990*). More recently, however, it has become clear that adult stem cells are not universally quiescent but instead cycle in accordance with the needs of the tissue: Some types of stem cells proliferate continuously, whereas others switch from quiescence to rapid proliferation in response to certain stimuli (e.g. wounding or hormones) (*Wabik and Jones, 2015*). In mammals, for example, hematopoietic and neural stem cells reversibly switch between quiescence and active proliferation in response to tissue injury (*Doetsch et al., 1999*; *Harrison and Lerner, 1991*; *Lugert et al., 2010*), and mammary stem cells expand transiently during pregnancy and the estrus cycle (*Asselin-Labat et al., 2010*; *Joshi et al., 2010*). Though periods of sustained stem cell proliferation enable rapid tissue growth or turnover, they challenge the view of quiescence as a prerequisite for the stem cell fate. Thus, a long-standing question has remained unanswered: Does cell-cycle quiescence play a role in stem cell maintenance?

Understanding the relationship between cell-cycle quiescence and stem cell maintenance has been difficult because tractable models of facultative stem cell quiescence have been lacking. Perturbations affecting the cell cycle can in some cases impact stem cell maintenance (*Orford and Scadden, 2008*; *Pietras et al., 2011*; *Yilmaz et al., 2012*), but whether quiescence can maintain stem cells independent of the signals otherwise required for their maintenance has been untested. Such a test requires a system in which cell-cycle quiescence can be readily induced, and in which the signals otherwise required for stem cell maintenance can be readily removed. In this study, we establish the

**eLife digest** Adult stem cells can divide to produce cells that can develop into one of many different specialist cell types in a tissue, and so are vitally important for tissue repair and maintenance. Some types of adult stem cells exist primarily in a non-dividing state known as quiescence, which for a long time was thought to be essential for maintaining the stem cell state. However, researchers have discovered some adult stem cells that are either not quiescent, or only enter this state rarely.

Until now, biologists have lacked an experimental model in which the role of quiescence in maintaining stem cells can be easily investigated. Seidel and Kimble have now investigated the role of quiescence in the germline stem cells – which give rise to egg and sperm cells – of the roundworm *Caenorhabditis elegans*. The results of the study revealed that although the germline stem cells divide continuously when the worms are well fed, starving the worms causes these stem cells to become quiescent.

Maintaining *C. elegans* germline stem cells in a stem cell state normally involves a process called Notch signaling, which cells use to communicate with each other. However, Seidel and Kimble found that the germline quiescence caused by starvation maintains the stem cell state even when Notch signaling is prevented. This suggests that, in the absence of food, quiescence alone can maintain germline stem cells, although how it does so remains a question for future work. One possibility is that quiescence stabilizes other molecules involved in the Notch signaling pathway or prevents the production of proteins that enable a stem cell to develop into a specialized cell.

adult germline of *Caenorhabditis elegans* as a model fitting these criteria. We describe a previously uncharacterized state of cell-cycle quiescence among adult germline stem cells, emerging under conditions of starvation. We then test whether this quiescence can maintain stem cells, independent of the signal required for their maintenance under conditions of active proliferation.

The adult germline of *C. elegans* presents a tractable model for studying stem cell behavior because of its simple, linear organization (*Figure 1A*). Mitotically dividing germ cells—including germline stem cells—reside in the distal region of the gonad (the 'progenitor zone'). Differentiating germ cells, in meiotic prophase, are located more proximally. (Here, we use the term 'progenitor zone' rather than the earlier term 'mitotic zone' or 'proliferative zone' to reflect the facultative nature of germ cell divisions.) The progenitor zone has been studied under fed conditions and is composed of a distal pool of germline stem cells and a more proximal pool of cells that have begun to differentiate (*Cinquin et al., 2010*). This proximal pool comprises cells dividing mitotically, as well as cells completing their final passage through interphase in preparation for entry into the meiotic cell cycle. We collectively refer to these cells as 'transient progenitors', to reflect their continued mitotic divisions and transitional state (*Figure 1A*). Under fed conditions, cells throughout the progenitor zone cycle asynchronously and continuously (*Crittenden et al., 2006*; *Fox et al., 2011*; *Jaramillo-Lambert et al., 2007*; *Morgan et al., 2010*), with transient progenitors undergoing one or two rounds of division as they pass through the proximal progenitor zone (*Fox and Schedl, 2015*).

Prior to this work, germ cell proliferation in *C. elegans* adults had not been examined in detail under food-limited conditions. However, the effects of such conditions have been examined during larval development in *C. elegans*, as well as in adult *Drosophila*, and in both contexts, germ cells respond robustly to nutritional cues. In *Drosophila*, nutrient limitation or changes in nutrient-sensing pathways slow germ cell proliferation, reduce germline stem cell number, or both (*Armstrong et al., 2014*; *Drummond-Barbosa and Spradling, 2001*; *Hsu et al., 2008*; *LaFever et al., 2010*; *McLeod et al., 2010*; *Roth et al., 2012*; *Sheng and Matunis, 2011*). These effects are mediated in part by changes in the somatic gonad (*Yang and Yamashita, 2015*), including changes in the size of the somatic niche supporting germline stem cells (*Bonfini et al., 2015*; *Hsu and Drummond-Barbosa, 2011*). In *C. elegans*, primordial germ cells are born in the early embryo and arrest in the G2 phase of the cell cycle until newly hatched larvae begin to feed (*Butuci et al., 2015*; *Fukuyama et al., 2006*; *Fukuyama et al., 2012*). This response to feeding has been hypothesized to involve food-related signals traveling through soma-to-germline gap junctions, which are required early in larval development for germ cell proliferation and survival (*Starich et al., 2014*). Later in

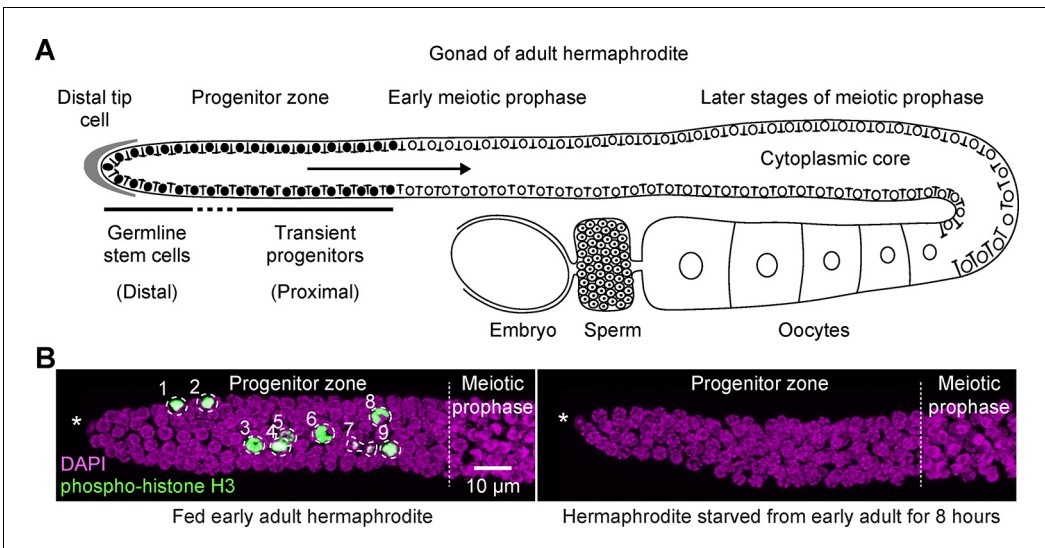

**Figure 1.** Fed versus starved adult hermaphrodite gonad of *Caenorhabditis elegans*. (**A**) Schematic of an adult hermaphrodite gonadal arm, with the progenitor zone at its distal end and maturing gametes at its proximal end. Germline stem cells and transient progenitors are located in the distal and proximal progenitor zone, respectively. Cells in both pools cycle asynchronously, although they are partially connected via a cytoplasmic core. Filled circles, germ cell nuclei in the progenitor zone. Open circles, germ cell nuclei in meiotic prophase, including developing oocytes. Gonads in males and larval hermaphrodites are organized similarly, although their proximal germ cells differentiate as sperm. This same gonad organization is also seen in starved animals of any stage or sex for time intervals examined in this work. (**B**) Images of distal gonads dissected from adult hermaphrodites and stained with DAPI to visualize DNA (magenta) and anti-phospho-histone H3 to visualize M-phase chromosomes (green). M-phase cells are outlined and numbered. Left, fed early adult hermaphrodite. Right, hermaphrodite starved from early adult for 8 hr. (See Materials and methods for definition of 'early adult'.) Asterisks, distal gonad ends. Images are maximum-intensity z-projections.

development, germ cells stop dividing if animals enter the non-feeding dauer larval stage (*Narbonne and Roy, 2006*). Even in non-dauer larvae, germ cells proliferate less when food is scarce, an effect mediated in part by communication between food-sensing neurons and the somatic gonad (*Dalfo et al., 2012*; *Korta et al., 2012*). In adult *C. elegans*, decreased food intake slows mitotic and meiotic progression and oogenesis (*Gerhold et al., 2015*; *Lopez et al., 2013*; *Salinas et al., 2006*; *Seidel and Kimble, 2011*), and limited observations suggest that germ cell proliferation is also reduced (*Salinas et al., 2006*). More strikingly, full starvation from the L4 larval stage causes dramatic germline shrinkage in adult hermaphrodites, and this shrinkage is reversible upon re-feeding (*Angelo and Van Gilst, 2009*; *Seidel and Kimble, 2011*). These observations motivated us to examine in greater detail how mitotically dividing germ cells in adult *C. elegans* respond to food removal.

Here, we report that in the absence of food, mitotically dividing germ cells in adult *C. elegans* stop dividing and become quiescent. This quiescence is characterized by a dramatic slowing of S phase, cell-cycle arrest in G2, and the ability to re-enter M phase rapidly in response to re-feeding. We investigate these cell-cycle responses in wildtype animals and in germline tumors, and we test whether this cell-cycle quiescence requires factors controlling larval or behavioral responses to food. We next investigate the control of stem cell maintenance under starved conditions. We uncover a major difference in the requirement for GLP-1/Notch signaling in the maintenance of actively proliferating versus quiescent germline stem cells. This work establishes the *C. elegans* germline as a model of facultative stem cell quiescence and demonstrates the utility of such a model in clarifying the role of quiescence in maintaining the stem cell state.

## Results

### M-phase entry in adult germ cells responds rapidly to starvation and re-feeding

To investigate how starvation affects germ cell division in adults, we removed food from early adult hermaphrodites and males and monitored the number of germ cells in M phase over the following 10.5 hr. Cells in M phase were identified by staining for phospho-histone H3 (*Figure 1B*), a marker of M phase (*Hans and Dimitrov, 2001*). Food removal caused a drop in the number of M-phase cells (*Figure 2A*), and this response was fast: In hermaphrodites, the number of M-phase cells per progenitor zone dropped from an average of 7.6 before food removal to 2.1 after 30 min without food (n = 7 replicates of 220–551 gonadal arms per replicate per time point) (*Figure 2A*). The number of M-phase cells continued to decline thereafter, and after 3.5 hr without food, M-phase cells were virtually absent (*Figure 2A*). This drop in M-phase cells did not occur in hermaphrodites fed

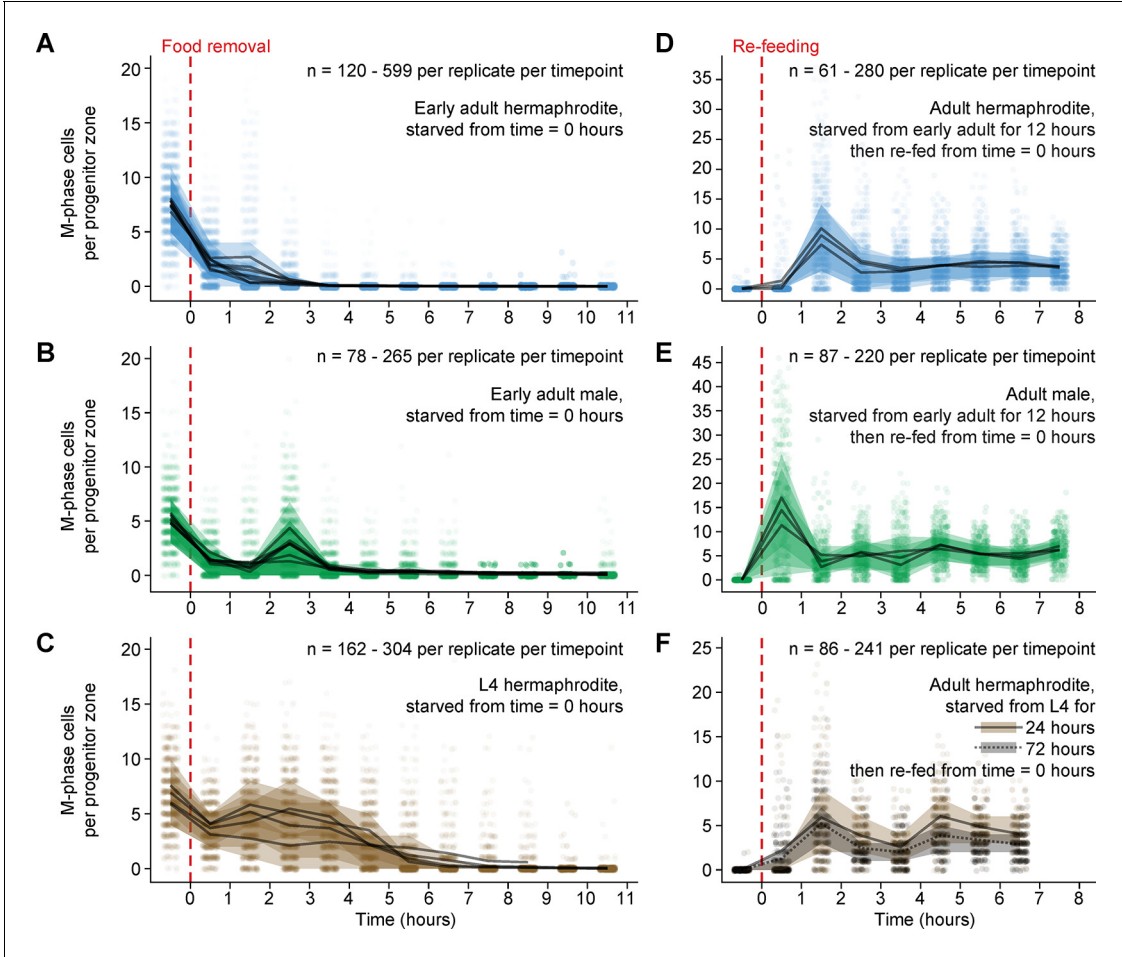

**Figure 2.** Mitotic divisions in adult progenitor zones respond quickly to food removal and re-feeding. Time courses showing the number M-phase cells per progenitor zone after food removal or re-feeding. Time zero indicates the start of food removal or re-feeding. Animals in **A**, **B** and **D**, **E** were starved from early adult. Animals in **C** were starved from mid L4. Animals in **F** were starved from mid L4 for 24 hr or from early L4 for 72 hr. Independent replicates are overplotted with transparency. For each replicate, lines connect means, and shaded areas show interquartile ranges. Sample sizes indicate numbers of gonadal arms. Source data are available in *Figure 2—source data 1*.

The following source data and figure supplement are available for figure 2:

**Source data 1.** Counts of M-phase cells for starvation and re-feeding time courses of wildtype animals.

**Figure supplement 1.** Comparison of numbers of M-phase cells in fed, starved, and re-fed animals.

continuously (*Figure 2—figure supplement 1B, D*), nor in hermaphrodites exposed to a mock starvation procedure (*Figure 2—figure supplement 1C*). In males, M-phase cells also disappeared rapidly in response to food removal, although the initial drop in M-phase cells was not monotonically decreasing (*Figure 2B*). We conclude that in adults of both sexes, germ cells stop dividing quickly in the absence of food.

We next investigated how germ cell division responds to re-feeding. We removed food from early adult hermaphrodites and males, allowed animals to remain in starvation for 12 hr, then re-fed animals, and monitored the number of M-phase cells, as above. In hermaphrodites, this treatment triggered a burst of M-phase cells 1.5 hr after the start of re-feeding (*Figure 2D*). Males showed a similar response to re-feeding, but the burst of M-phase cells occurred 1 hr earlier (*Figure 2E*). In both sexes, these bursts included some individual germlines having approximately twice as many M-phase cells as were observed among continuously fed animals (*Figure 2—figure supplement 1E, F*). These results demonstrate that in both sexes, germ cells resume mitotic division rapidly in response to re-feeding. The faster response in males is consistent with germ cells in males having a faster cell-cycle under continuously fed conditions (*Morgan et al., 2010*). Further, the higher maxima of M-phase cells in re-fed versus continuously fed animals is consistent with germ cells collecting at the G2-to-M transition during starvation and entering M phase semi-synchronously upon re-feeding.

## Cessation of M-phase entry in response to starvation coincides with the molt into adulthood

We next extended our results to adult hermaphrodites starved from the L4 larval stage. This extension was motivated by the need to perform certain later experiments in such animals, as starvation from L4 prolongs the amount of time that adult hermaphrodites can be maintained without food (*Angelo and Van Gilst, 2009*; *Seidel and Kimble, 2011*). We removed food from mid-L4 hermaphrodites and monitored the number of cells in M phase, as above. In starved L4s, M-phase cells persisted for ~4–5 hr after food removal, with the average number of M-phase cells only moderately reduced relative to fed animals (*Figure 2C*). Thereafter, the number of M-phase cells declined rapidly, and after 10.5 hr without food, M-phase cells had virtually disappeared (*Figure 2C*). The disappearance of M-phase cells coincided with the molt into adulthood (~5–8 hr after food removal), and the coincidence of these events persisted even under conditions where the timing of this molt was changed: Hermaphrodites starved from early L4 molted into adulthood ~12–20 hr after food removal, and gonads in these animals contained, on average, 1.6 M-phase cells per progenitor zone before the molt (n = 57, gonads collected 10.5 hr post food removal) and 0.0 M-phase cells after the molt (n = 63, gonads collected 24 hr post food removal). These results demonstrate that germ cells in hermaphrodites starved from L4 do not immediately stop dividing in response to food removal, but germ cell division eventually ceases, at or near the molt into adulthood. This finding suggests that mitotically dividing germ cells in L4s are not equivalent to those in adults, a result consistent with previous studies (*Crittenden et al., 2002*; *Dalfo et al., 2012*; *Gerhold et al., 2015*; *Michaelson et al., 2010*).

## Longer starvation does not delay M-phase entry upon re-feeding

In other systems, re-entry into the mitotic cell cycle following a period of quiescence occurs more slowly after longer periods of quiescence (*Lum et al., 2005*; *Soprano, 1994*). We therefore tested whether longer periods of starvation would delay mitotic re-entry upon re-feeding. We repeated the re-feeding time course in two types of animals having experienced longer starvation: Adult hermaphrodites starved from mid-L4 for 24 hr and adult hermaphrodites starved from early L4 for 72 hr. In both types of animals, re-feeding triggered a burst of M-phase cells 1.5 hr after re-feeding (*Figure 2F*), similar to the re-feeding response in animals starved for only 12 hr (compare *Figure 2D* versus F) . We conclude that the timing of M-phase entry upon re-feeding is largely unaffected by the duration of preceding starvation, at least during the first 72 hr of starvation.

## During starvation, germ cells progress slowly through S phase and arrest in G2

We next examined how starvation affects progression of germ cells through G1, S phase, and G2. First, we monitored cell-cycle progression in fed animals. By labeling germlines with the thymidine

analog 5-ethynyl-2′-deoxyuridine (EdU) and monitoring the fraction of EdU⁺ M-phase cells over time, we estimated a median cell-cycle length in fed early adult hermaphrodites of ~6.2 hr, with S phase lasting ~4.4 hr, G2 lasting ~1.3 hr, and G1 and M phase together lasting ~30 min (*Figure 3— figure supplement 1*). We also measured cell-cycle length in fed hermaphrodites aged 24-hr post mid-L4 (~12–16 hr past the early adult stage). For this age group, we estimated a median cell-cycle length of ~9.8 hr, with median G1, S-phase, and G2 lengths of <30 min, ~6.8 hr, and ~2.3 hr, respectively, but with a long-tailed distribution of G2 lengths (*Figure 3—figure supplement 1*). Our estimates of median cell-cycle length are within the range reported by others (*Crittenden et al., 2006*; *Fox et al., 2011*; *Jaramillo-Lambert et al., 2007*; *Morgan et al., 2010*), and the long-tailed distribution of G2 lengths is consistent with estimates of maximal cell-cycle length being considerably longer than estimates of median cell-cycle length (*Crittenden et al., 2006*; *Fox et al., 2011*; *Jaramillo-Lambert et al., 2007*; *Morgan et al., 2010*).

Second, we monitored cell-cycle progression during starvation. We removed food from animals of both sexes and, after varying amounts of time, calculated the fraction of progenitor-zone cells in G1, S phase, and G2. S-phase cells were identified by EdU labeling, and G1 versus G2 cells were distinguished by nuclear size (see 'Use of nuclear size to distinguish G1 versus G2 cells' in Materials and methods and *Figure 3—figure supplement 2*). The following time points were collected: Animals starved from early adult for 3.5, 6.5, and 10.5 hr; animals starved from mid-L4 for 10.5 and 24 hr; and animals starved from early L4 for 10.5, 24, 48, and 72 hr. For all three age groups and for both sexes, G1 cells disappeared following food removal, and the timing of their disappearance coincided with the disappearance of M-phase cells (*Figure 3A*). This result demonstrates that during starvation, G1 cells continued to initiate S phase without a pronounced delay. Our second observation was that for all age groups and sexes, the fraction of S-phase cells decreased over time during starvation, and the fraction of G2 cells increased (*Figure 3A*). However, S-phase and G2 fractions changed more slowly during starvation than expected from measurements of S-phase length in fed animals (*Figure 3A*). (For example, in hermaphrodites starved from early adult, S-phase fractions changed very little during the 7 hr between time points 3.5 hr and 10.5 hr [*Figure 3A*], indicating that progression through S phase during this time period was minimal. By contrast, cells in fed adult hermaphrodites complete S phase in a median of ~4.4–6.8 hr [*Figure 3—figure supplement 1*].) We conclude that during starvation, germ cells do not pause in G1 and instead continue through S phase and arrest in G2. However, S-phase progression occurs much more slowly than under fed conditions.

## Re-feeding restores the rate of progression through S phase and G2

We next examined the effect of re-feeding on cell-cycle progression through S phase and G2. We re-fed starved animals and determined the length of time until all germ cells entered M phase. This analysis is complementary to our initial monitoring of M-phase cells (*Figure 2*) because our initial monitoring allowed us to detect the earliest germ cells to enter M-phase upon re-feeding, whereas this analysis allowed us to detect the slowest such cells. To detect M-phase entry of all cells, we blocked M-phase exit using a temperature-sensitive mutation in *emb-30*, which encodes a component of the anaphase-promoting complex (*Furuta et al., 2000*). *emb-30(tn377ts)* hermaphrodites were starved at the permissive temperature (15°C), then shifted to the restrictive temperature (25°C), and re-fed. At the time of re-feeding, germlines contained a mixture of S-phase and G2 cells (data not shown). In response to re-feeding, germ cells entered M phase, as expected, but were unable to complete the metaphase-to-anaphase transition, thus enabling us to quantify the accumulation of M-phase cells: Two hours after re-feeding, 55% of cells, on average, had entered M phase (n = 60 gonadal arms); after 4 hr, this number reached 99% (n = 66 gonadal arms) (*Figure 4B*). (This analysis is restricted to the distal-most 50 germ cells, because germ cells located more proximally sometimes directly entered the meiotic cell cycle upon re-feeding. See *Figure 4*.) Therefore, virtually all mitotically dividing germ cells completed the remainder of S phase and G2 within 4 hr of re-feeding, a time shorter than the time required to complete all of S phase and G2 in continuously fed adult hermaphrodites (~4.4–6.8 hr for S-phase, plus ~1.3–2.3 hr for G2). Thus, not only does re-feeding trigger germ cells arrested in G2 to enter M phase, but re-feeding also restores the rate of progression through S phase and earlier stages of G2.

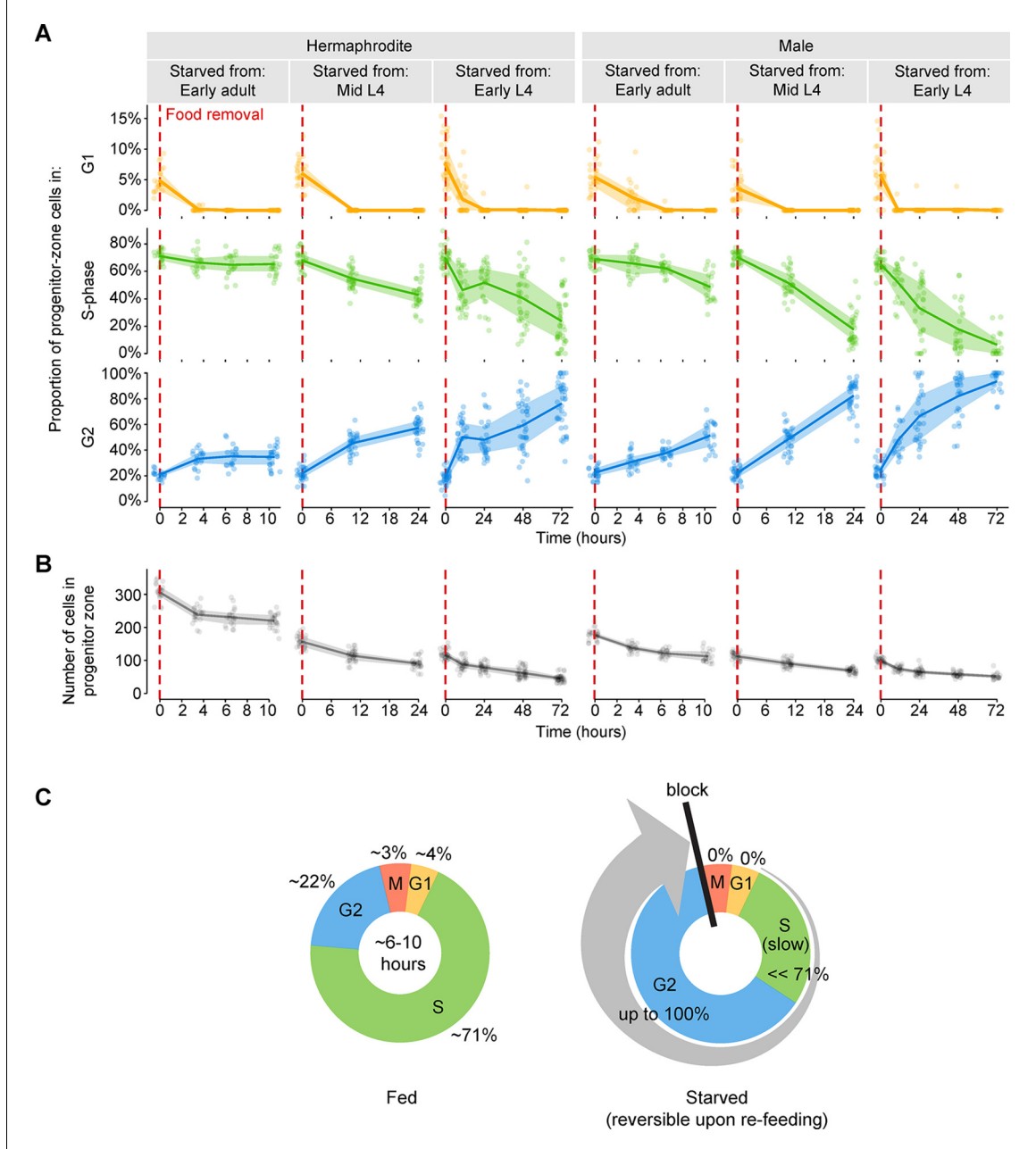

**Figure 3.** Starvation slows S-phase and causes germ cells to arrest in G2. (**A**) Time courses showing the proportion of progenitor-zone cells in G1, S-phase, or G2, for animals starved from early adult or from mid or early L4. Animals starved from mid L4 were adults at the 10.5- and 24-hr time points. Animals starved from early L4 were adults at the 24-, 48-, and 72-hr time points. n = 19–40 gonadal arms per time point. (**B**) Time courses showing the number of cells in the progenitor zone, for the same gonads used in A. Within each plot, lines connect means, and shaded areas show interquartile ranges. (**C**) Schematic summarizing the effect of starvation on the mitotic cell cycle of germ cells. Cell-cycle length under fed conditions was measured in adult hermaphrodites (*Figure 3—figure supplement 1*). Source data are available in *Figure 3—source data 1*.

The following source data and figure supplements are available for figure 3:

**Source data 1.** Counts of cells in each phase of the cell cycle for starvation time courses of wildtype animals.

**Source data 2.** Counts of cells in each phase of the cell cycle for fed adult wildtype hermaphrodites.

**Source data 3.** EdU labeling in fed adult wildtype hermaphrodites.

*Figure 3. continued on next page*

*Figure 3. Continued*

**Source data 4.** Propidium iodide intensities versus nuclear volume in the progenitor zone.

**Figure supplement 1.** Measurements of cell-cycle length in fed animals.

**Figure supplement 2.** G1 versus G2 cells can be distinguished by nuclear size in EdU-labeled gonads.

## Starvation-induced quiescence does not require proximity to the germline stem cell niche

Mitotically dividing germ cells in adults are confined to the distal gonad, where they contact a single somatic cell—the distal tip cell (or pair of distal tip cells, in males) (*Figure 1A*). The distal tip cell forms the niche for germline stem cells (*Kimble and Seidel, 2013*) and influences how germ cells respond to physiological cues (*Dalfo et al., 2012*). We therefore tested the effect of proximity to the distal tip cell on cell-cycle responses to food removal and re-feeding. We monitored M-phase cells, as above, in two mutant backgrounds in which mitotically dividing germ cells fill the gonad (i.e. germline tumors): (i) *glp-1(oz112gf)/Notch* gain-of-function mutants, in which constitutive GLP-1/ Notch signaling maintains all germ cells in the mitotic cell cycle (*Berry et al., 1997*) and (ii) *gld-3 (q730) nos-3(q650)* loss-of-function mutants, in which meiotic entry is inhibited irrespective of GLP-1/ Notch signaling (*Byrd et al., 2014*; *Eckmann et al., 2004*). Germ cells in both genotypes of germline tumors responded normally to starvation and re-feeding: Outside the region normally corresponding to the progenitor zone (i.e. outside the distal-most 20 rows of germ cells), M-phase cells disappeared quickly in response to food removal and re-bounded 1 to 2 hr after re-feeding (*Figure 5*). These results demonstrate that proximity to the distal tip cell—the germline stem cell niche—is not required for a normal starvation and re-feeding response. Additionally, these results refine our understanding of the control of germ cell fate: Constitutive GLP-1/Notch signaling or combined loss of *gld-3* and *nos-3* does not promote germ cell proliferation per se, but rather promotes an undifferentiated fate in which cells divide only in the presence of food.

## Starvation-induced quiescence maintains germline stem cells independent of GLP-1/Notch

In multiple types of stem and progenitor cells, cell-cycle quiescence correlates with the capacity for long-term self-renewal (*Cheung and Rando, 2013*; *Orford and Scadden, 2008*). We therefore investigated how quiescence affects the maintenance of *C. elegans* germline stem cells. Under fed conditions, maintenance of these stem cells requires GLP-1/Notch signaling (*Austin and Kimble, 1987*). The *glp-1* gene, which encodes one of two Notch receptors in *C. elegans*, is expressed in the progenitor zone, and the receptor is activated by ligands expressed in the adjacent distal tip cell (*Henderson et al., 1994*; *Kimble and Crittenden, 2007*; *Nadarajan et al., 2009*). We used the temperature-sensitive *glp-1* allele *q224ts* to test whether GLP-1/Notch signaling is similarly required for maintenance of germline stem cells under starved conditions. The *q224ts* allele is the strongest of all known temperature-sensitive *glp-1* alleles and behaves like a null at the restrictive temperature (*Austin and Kimble, 1987*; *Kodoyianni et al., 1992*).

In fed animals, removal of GLP-1/Notch signaling causes germline stem cells to be lost: Germline stem cells fail to self-renew, and instead all germ cells enter the meiotic cell cycle (*Austin and Kimble, 1987*; *Cinquin et al., 2010*; *Fox and Schedl, 2015*). We tested whether the loss of GLP-1/Notch signaling produces these same effects in starved animals. We removed food from *glp-1(q224ts)* hermaphrodites at the permissive temperature (15°C), shifted starved animals to the restrictive temperature (25°C) for 8 hr, and then evaluated germ cell fate. Cell fate was assessed by staining for the meiosis-associated protein GLD-1 (*Jones et al., 1996*) and by scoring cells for the 'crescent' chromosome morphology indicative of meiotic chromosome pairing (*Dernburg et al., 1998*). This morphology is readily distinguishable from the 'non-crescent' morphology found in mitotic interphase. In fed controls, incubation at the restrictive temperature caused all germ cells to enter the meiotic cell cycle: Chromosomes adopted the 'crescent' shape (*Figure 6A*), and GLD-1 levels in the distal-most germ cells rose (*Figure 6I*). In starved animals, by contrast, meiotic entry did not occur: 99% (n = 214) of progenitor zones retained germ cells with a interphase chromosome

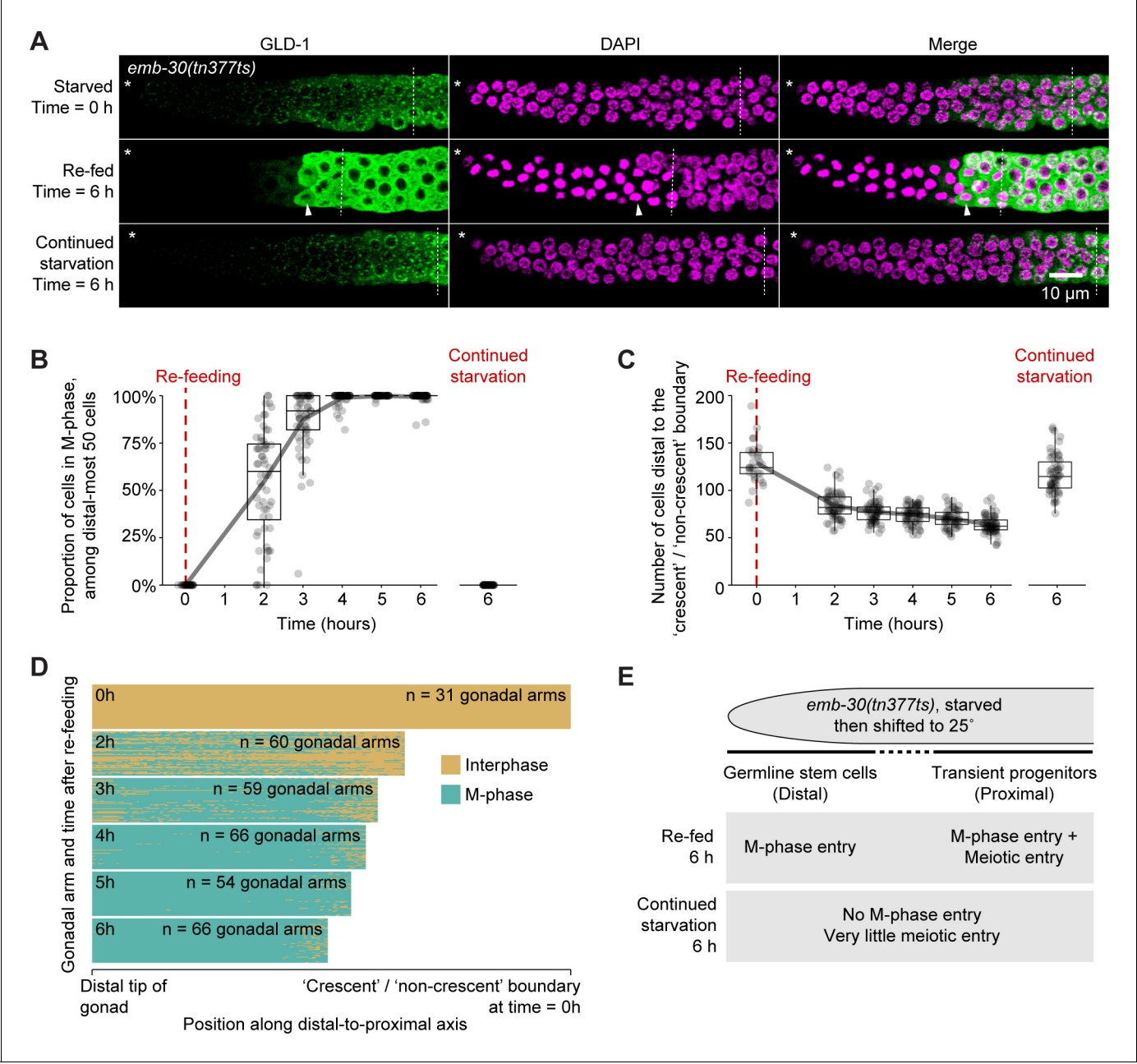

**Figure 4.** Re-feeding restores the rate of progression through S phase and G2, as well as the meiotic entry of transient progenitors. (**A**) Images of distal gonads dissected from *emb-30(tn377ts)* adult hermaphrodites and stained with DAPI to visualize DNA (magenta) and anti-GLD-1 (green). Animals were starved at 15°C, shifted to 25°C, and then either re-fed or maintained in starvation. Time = 0 hr indicates the time at which re-feeding was started or starvation continued. Top, starved adult hermaphrodite, Time = 0 hr. Middle, re-fed adult hermaphrodite, Time = 6 hr. Bottom, adult hermaphrodite maintained in starvation, Time = 6 hr. Dashed lines, 'crescent'/'non-crescent' boundaries (defined by the second distal-most 'crescent' cell—see Materials and methods section). Arrowhead, an example of a metaphase-arrested cell having a high level of GLD-1. Asterisks, distal gonad ends. Images are maximum-intensity z-projections. (**B**) Time course showing the proportion of cells in M-phase (among the distal-most 50 germ cells) for *emb-30(tn377ts)* adult hermaphrodites treated as in A. n = 31–66 gonadal arms per time point. (**C**) Number of cells distal to the 'crescent'/'non-crescent' boundary for the same gonads used in B. (**D**) Cell-cycle behavior, among cells distal to the 'crescent'/'non-crescent' boundary, for the same gonads used in B. Cells from individual gonadal arms are plotted along lines, with color indicating cell-cycle phase (interphase or M-phase) and location determined by each cell's position along the distal-to-proximal axis of distal gonad. For each time point, this axis is scaled relative to the mean number of cells distal to the 'crescent'/'non-crescent' boundary. (**E**) Summary of the effects of re-feeding and continued starvation on germ cells in *emb-30(tn377ts)* hermaphrodites in the distal versus proximal progenitor zone. Source data are available in *Figure 4—source data 1*.

*Figure 4. continued on next page*

*Figure 4. Continued*

The following source data is available for figure 4:

**Source data 1.** Temperature-shift experiments of *emb-30(tn377ts)* hermaphrodites.

morphology (*Figure 6C*), and GLD-1 levels in the distal-most germ cells remained low (n = 47 of 47 gonadal arms) (*Figure 6I*). Importantly, germ cells in starved animals retained the capacity for mitotic cell division, because when starved animals were re-fed at the restrictive temperature, their germ cells re-entered M phase (*Figure 6G*). Germ cells in starved animals also retained the capacity for long-term self-renewal, because when starved animals were instead returned to the permissive temperature and re-fed for 2–3 days, 91% (n = 148) of progenitor zones retained germ cells in the mitotic cell cycle (*Figure 6C*). Similar results were observed when incubation at the restrictive temperature was extended to 16 hr or 24 hr (*Figure 6D*). These results demonstrate that starved animals maintain germline stem cells independent of GLP-1/Notch. In other words, starvation inhibits the meiotic entry of germline stem cells, even in the absence of GLP-1/Notch. Similar to cell-cycle quiescence, this inhibition of meiotic entry was reversible upon re-feeding, because when starved *glp-1 (q224ts)* animals were re-fed at the restrictive temperature, all germ cells eventually entered the meiotic cell cycle (*Figure 6H*).

## Quiescence induced by non-starvation conditions maintains germline stem cells independent of GLP-1/Notch

To further investigate the relationship between cell-cycle quiescence and stem cell maintenance, we asked, apart from starvation, do other conditions that inhibit germ cell division also maintain germline stem cells independent of GLP-1/Notch? To answer this question, we performed temperature-shift experiments in *glp-1(q224ts)* hermaphrodites under two additional conditions: High NaCl and absence of sperm, each of which causes a twofold drop in the germ cell mitotic index (*Morgan et al., 2010*; *Salinas et al., 2006*). Animals exposed to high NaCl (300 mM) or lacking sperm (via loss-of-function mutation in *fog-1*) were grown at the permissive temperature, shifted to the restrictive temperature for 8 hr, and then returned to the permissive temperature for 2–3 days. Following this treatment, germ cell fate (mitotic versus meiotic) was evaluated by scoring germ cells for the 'crescent' chromosome morphology indicative of meiotic prophase (described above). For both high NaCl and absence of sperm, 48–79% (n = 129–231) of gonadal arms retained germ cells in the mitotic cell cycle (*Figure 6E–F*). To control for possible pleiotropic effects of the mutation used to eliminate sperm (*q785*), sperm was introduced to *fog-1(q785); glp-1(q224ts)* animals by mating with wildtype males, and these animals were examined in parallel. In such animals, incubation at the restrictive temperature caused all germ cells to enter the meiotic cell cycle (data not shown; n = 37 gonadal arms, scored immediately following incubation at the restrictive temperature). Thus, three stress conditions that inhibit or reduce germ cell division (starvation, high NaCl, and absence of sperm) also permitted maintenance of germline stem cells independent of GLP-1/Notch. This commonality suggests that cell-cycle quiescence itself is an effector of stem cell maintenance (*Figure 6J*). Nonetheless, such maintenance is not simply a function of inhibiting passage through M-phase, because stem cell maintenance in the absence of GLP-1/Notch was not permitted by cell-cycle arrest caused by RNAi knockdown of cyclin-dependent kinase 1 (*cdk-1*) (*Figure 6B*), a result we confirmed by visualizing formation of the synaptonemal complex (*Figure 6—figure supplement 1*).

## Starvation inhibits the meiotic entry of transient progenitors

The results above demonstrate that starvation inhibits the meiotic entry of germline stem cells, even in the absence of GLP-1/Notch. We therefore hypothesized that starvation might also control the meiotic entry of transient progenitors, located in the proximal progenitor zone (*Figure 1A*). No markers currently exist to distinguish transient progenitors from germline stem cells, and the boundary between these pools of cells is not clear, but the two pools can be distinguished under fed conditions by restricting movement of cells out of the progenitor zone (*Cinquin et al., 2010*). Under such conditions, transient progenitors enter the meiotic cell cycle, whereas germline stem cells do not (*Cinquin et al., 2010*). To compare meiotic entry of transient progenitors under fed versus starved conditions, we re-fed starved animals and restricted cell movement during re-feeding; we

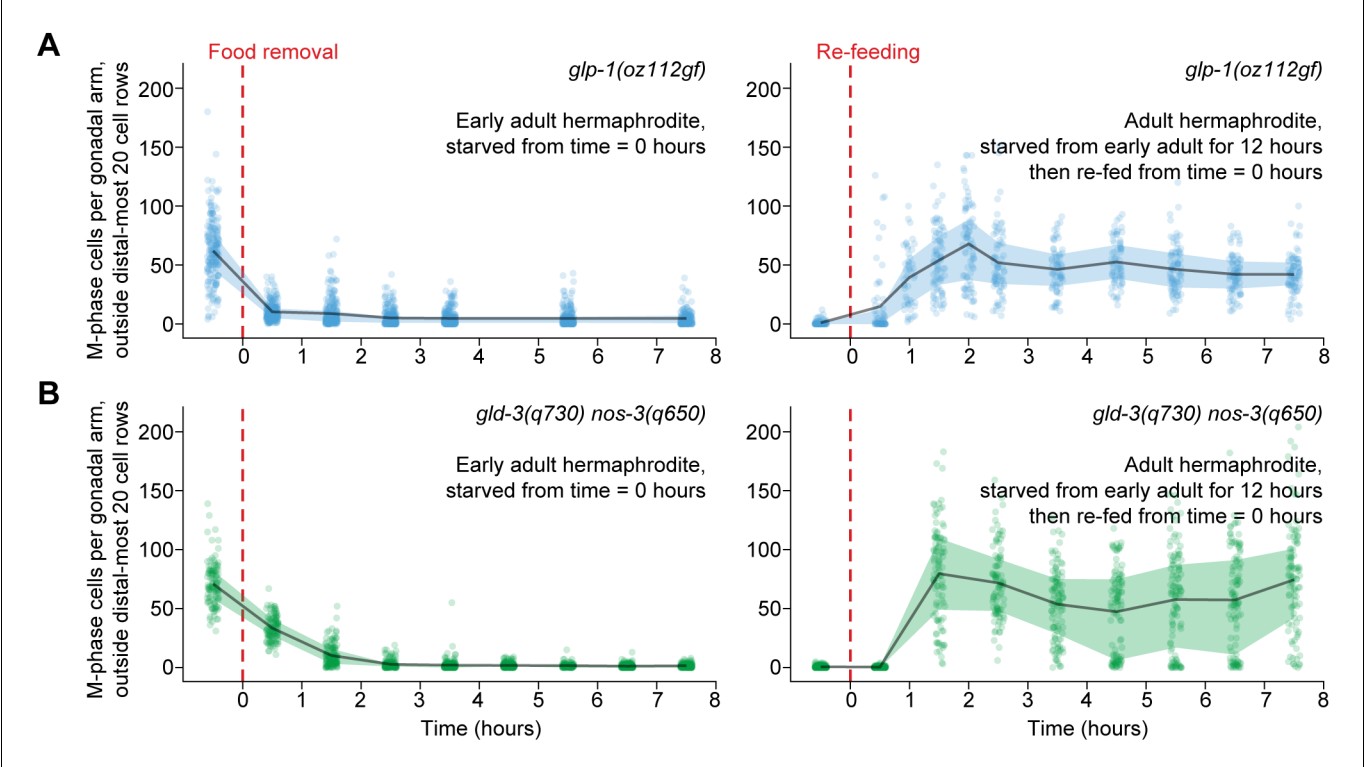

**Figure 5.** Ectopic mitotic divisions outside the progenitor zone respond normally to food removal and re-feeding. Time courses showing the number of M-phase cells—outside the distal-most 20 rows of germ cells—after food removal or re-feeding in adult *glp-1(oz112gf)* hermaphrodites or adult *gld-3 (q730) nos-3(q650)* hermaphrodites. Time zero indicates the start of food removal or re-feeding. Lines connect means, and shaded areas show interquartile ranges. n = 50–249 gonadal arms per time point. Source data are available in *Figure 5—source data 1*.

The following source data is available for figure 5:

**Source data 1.** Counts of M-phase cells for starvation and re-feeding time courses of *glp-1(oz112gf)* and *gld-3(q730) nos-3(q650)* hermaphrodites.

then compared meiotic entry after re-feeding to meiotic entry after continued starvation. Meiotic entry was assessed by staining for GLD-1 and by scoring cells for the 'crescent' chromosome morphology indicative of meiotic prophase (described above). Cell movement was restricted by performing this experiment in *emb-30(tn377ts)* hermaphrodites at the restrictive temperature (25°C), a condition that induces metaphase arrest (also described above, under 'Re-feeding restores the rate of progression through S phase and G2') (*Furuta et al., 2000*). *emb-30(tn377ts)* animals were starved at the permissive temperature (15°C), shifted to the restrictive temperature (25°C), and then re-fed for 6 hr or maintained in starvation. Our primary result from this experiment was that re-feeding and continued starvation affected meiotic entry differently. After 6 hr of re-feeding, cells in the proximal (but not distal) progenitor zone entered the meiotic cell cycle: GLD-1 levels in the proximal half of the progenitor zone rose (*Figure 4A*), and the boundary between 'crescent' and 'non-crescent' germ cells moved distally, such that the number of cells distal to this boundary was reduced by about half (*Figure 4C*). After 6 hr of continued starvation, by contrast, very little meiotic entry was observed: GLD-1 levels in the proximal progenitor zone remained largely unchanged (*Figure 4A*), and the number of cells distal to the 'crescent'/'non-crescent' boundary was only slightly reduced (*Figure 4C*). Our inference from these results is that the proximal half of the progenitor zone was composed of transient progenitors at the start of re-feeding, and that 6 hr of re-feeding—but not 6 hr of continued starvation—allowed for their timely progression into the meiotic cell cycle. We conclude that starvation slows or blocks the meiotic entry of transient progenitors, similar to its effect on the meiotic entry of germline stem cells.

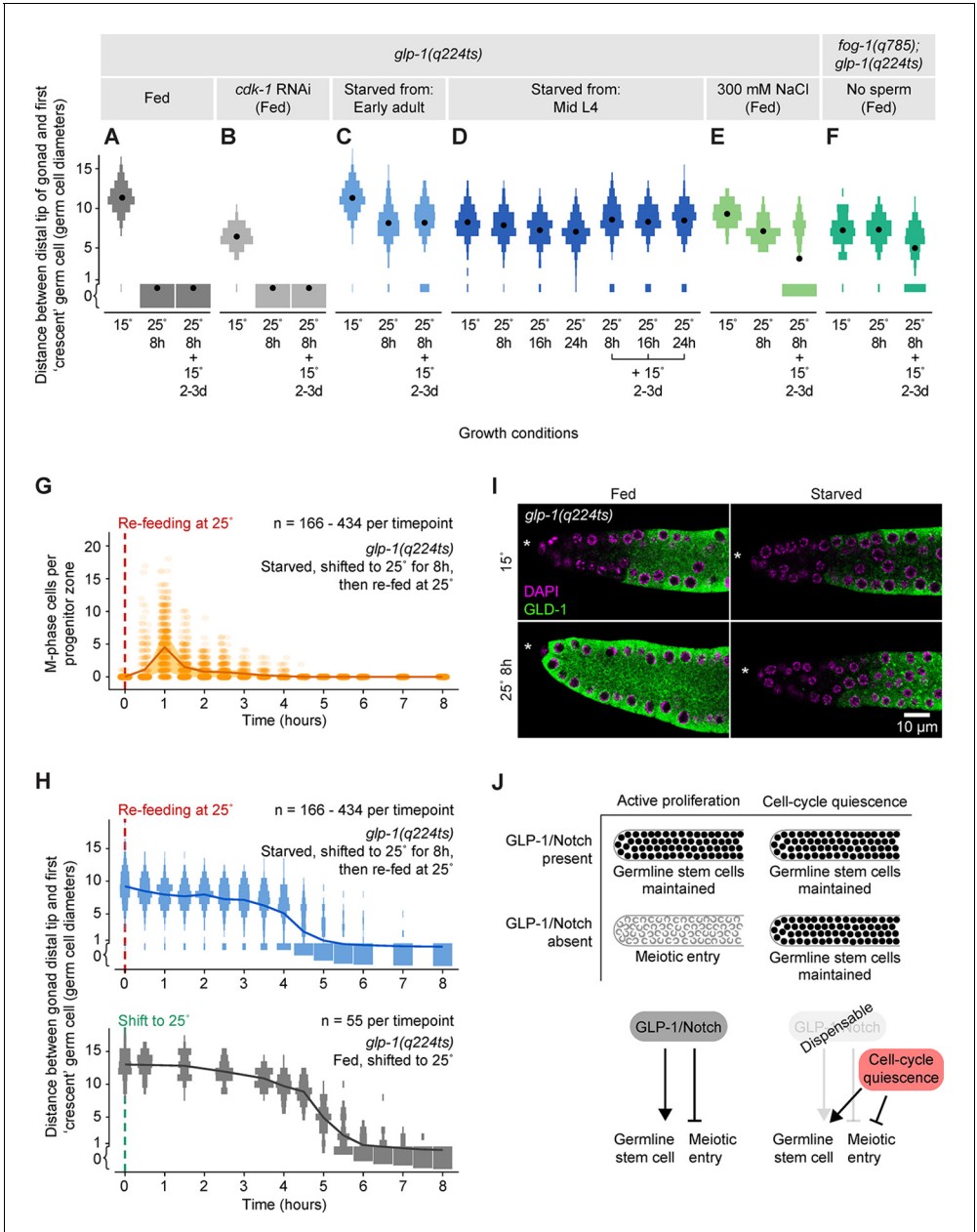

**Figure 6.** Quiescence induced by three different conditions maintains germline stem cells independent of GLP-1/Notch. (A–F) Distance between the gonad distal tip and the first 'crescent' germ cell for adult hermaphrodites of the genotypes and treatments shown. Animals were grown at 15°C, shifted to 25°C for 8, 16, or 24 hr, and then returned to 15°C for 2–3 days. Gonads were collected prior to the temperature shift (15°C), immediately following the temperature shift (25°C for 8 hr, 25°C for 16 hr, or 25°C for 24 hr), and following the 15°C recovery period (… + 15°C for 2–3 days). Data are plotted as vertical histograms, with black circles denoting means. n = 55–488 gonadal arms per time point. (G) Time course showing the number of M-phase cells per progenitor zone in *glp-1(q224ts)* hermaphrodites starved from early adult, shifted to 25°C for 8 hr, and then re-fed at 25°C. Lines connect means, and shaded areas show interquartile ranges. (H) Time courses showing distance between the gonad distal tip and the first 'crescent' germ cell. Top, hermaphrodites starved from early adult, shifted to 25°C for 8 hr, and then re-fed at 25°C. Bottom, hermaphrodites fed continuously and shifted to 25°C at the early adult stage. Data are plotted as vertical histograms, with lines connecting means. Sample sizes in G, H indicate number of gonadal arms. (I) Images of distal gonads dissected from adult hermaphrodites and stained with DAPI to visualize DNA (magenta) and anti-GLD-1 (green). Gonads were dissected before and after 8 hr at 25°C. Left-hand panels, fed adult hermaphrodites. Right-hand panels, hermaphrodites starved from early adult. Asterisks, distal gonad ends. (J) Schematic summarizing the effect of cell-cycle quiescence on germline stem cell maintenance. Under conditions of active proliferation, GLP-1/Notch is required for germline stem cell maintenance. Under quiescent conditions, GLP-1/Notch is dispensable. Source data for A–F are available in *Figure 6—source data 1*. Source data for G, H are available in *Figure 6—source data 2*.

*Figure 6. continued on next page*

*Figure 6. Continued*

The following source data and figure supplement are available for figure 6:

**Source data 1.** Temperature-shift experiments of *glp-1(q224)* hermaphrodites.
**Source data 2.** Time courses of temperature-shifted *glp-1(q224)* hermaphrodites.
**Figure supplement 1.** Cell-cycle arrest caused by *cdk-1* RNAi does not maintain germline stem cells in the absence of GLP-1/Notch.

## Starvation-induced quiescence is distinct from cell-cycle arrest induced by DNA damage

As a first step towards understanding the regulation of cell-cycle quiescence induced by starvation, we compared this quiescence to cell-cycle arrest caused by DNA damage, another perturbation causing G2 arrest (*Gartner et al., 2000*; *Kuntz and O'Connell, 2009*). Following three criteria, starvation-induced quiescence was distinct from cell-cycle arrest caused by DNA damage. First, DNA damage strongly up-regulates inhibitory phosphorylation of CDK-1 (*Figure 7A*; *Figure 7—figure supplement 1*; *Craig et al., 2012*). By contrast, starvation-induced quiescence did not up-regulate this phosphorylation (*Figure 7A*). Second, DNA damage causes germ cell nuclei to enlarge (*Figure 7A*; *Gartner et al., 2000*), a phenotype replicated by RNAi knockdown of *cdk-1* (*Jeong et al., 2011*). By contrast, starvation-induced quiescence did not cause nuclei to enlarge (*Figure 7A*). Third, cell-cycle arrest in response to DNA damage requires the p53 homolog *cep-1* (*Derry et al., 2007*). By contrast, starvation-induced quiescence did not require *cep-1* (*Figure 7B*). We conclude that starvation and DNA damage induce cell-cycle arrest differently.

## Starvation-induced quiescence does not require factors affecting larval and behavioral responses to food

As a second step towards understanding the regulation of starvation-induced quiescence, we tested whether this quiescence requires factors influencing the larval germline's response to food (*Dalfo et al., 2012*; *Michaelson et al., 2010*), including factors controlling germ cell quiescence during the two larval diapause states—L1 diapause (*Fukuyama et al., 2006*; *Fukuyama et al., 2012*) and the mid-larval dauer diapause (*Narbonne and Roy, 2006*). Food was removed from early adult hermaphrodites homozygous for mutations in the transforming growth factor beta (TGF-β) pathway, the insulin/insulin-like growth factor 1 (IGF-1) pathway, or the AMP-activated protein kinase (AMPK) pathway. M-phase cells were then monitored after food removal. Additionally, to test for a requirement for factors affecting behavioral responses to food, this same experiment was performed in animals defective for neuropeptide processing, neuropeptide secretion, or chemosensation, as well as in animals exposed to exogenous serotonin, which in some contexts acts as a food signal (*Luedtke et al., 2010*). In all experiments, M-phase cells disappeared quickly in response to food removal (*Figure 7B*). We conclude that each of the genes tested is not individually required for starvation-induced quiescence in adult germ cells; likewise, quiescence is not affected by exogenous serotonin. These findings suggest that quiescence in adults is controlled differently than cell-cycle responses in the larval germline and is largely independent of behavioral responses to food. These results are consistent with adult versus larval germ cells responding to food removal differently (*Figure 2A* versus C) and with reduced insulin/IGF-1 signaling or TGF-β signaling not affecting the germ cell mitotic index in fed adult hermaphrodites (*Dalfo et al., 2012*; *Michaelson et al., 2010*). Similarly, these results are consistent with no requirement for *daf-16/FOXO* in reducing the proliferation of *Drosophila* germline stem cells in response to poor diet (*Hsu et al., 2008*).

## Discussion

This work establishes the adult germline of *C. elegans* as a model of facultative stem cell quiescence in vivo and demonstrates that cell-cycle quiescence can maintain the stem cell fate, independent of a key signaling pathway otherwise required for this fate. Briefly, we find that in the absence of food, adult germ cells in *C. elegans* stop dividing and become quiescent (*Figure 8*). This quiescence is characterized by a slowing of S phase and a block to M-phase entry, such that cells arrest in the G2

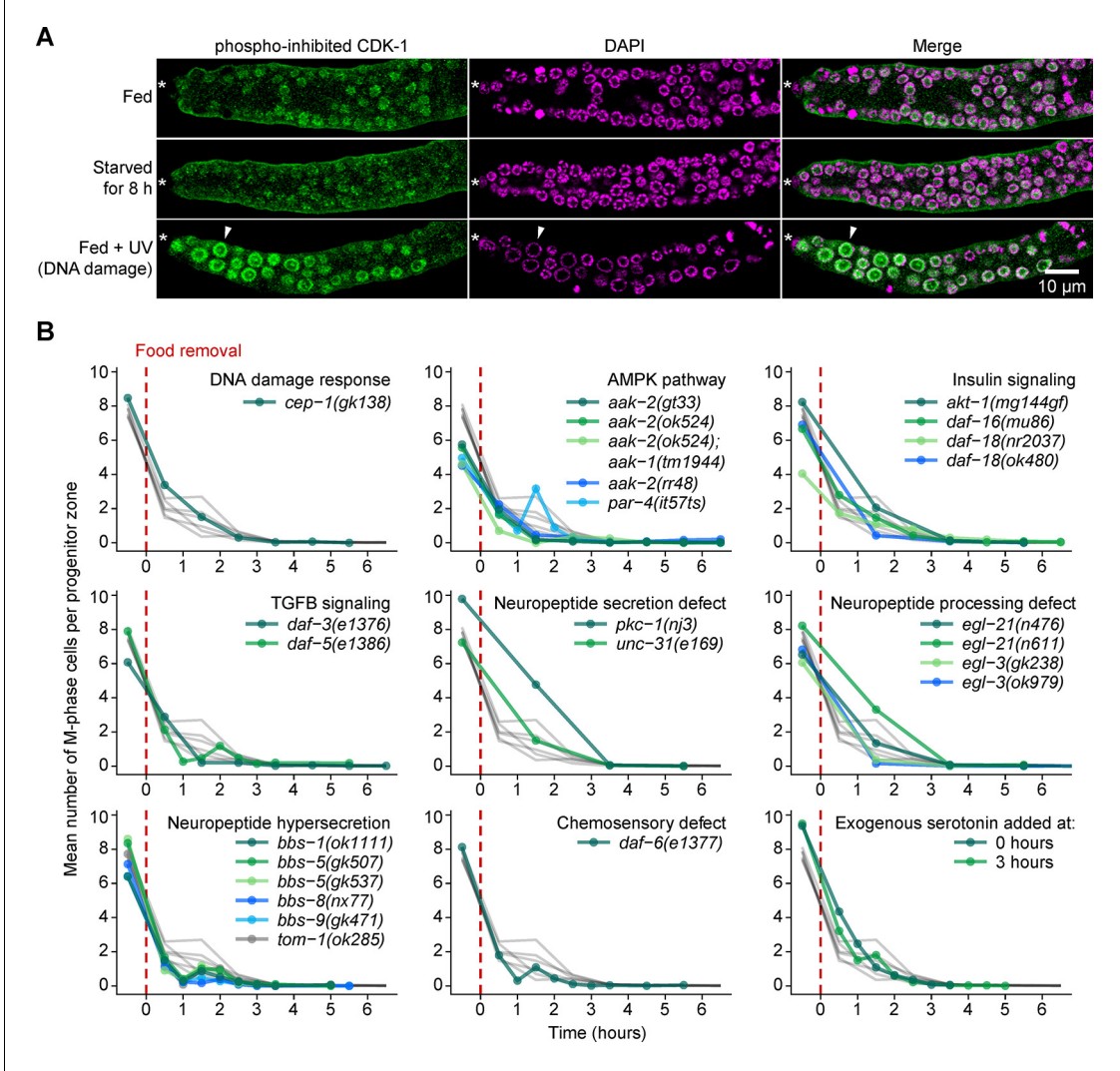

**Figure 7.** Starvation-induced quiescence is distinct from the DNA damage response and does not require factors involved in larval or behavioral responses to food. (A) Images of distal gonads dissected from adult hermaphrodites and stained with DAPI to visualize DNA (magenta) and anti-phospho-CDK-1 (green, Santa Cruz #sc-28435-R) to visualize inhibitory phosphorylation of CDK-1. The phospho-specificity of anti-phospho-CDK-1 is shown in *Figure 7—figure supplement 1*. Top row, fed adult hermaphrodite. Center row, adult hermaphrodite starved from early adult for 8 hr. Bottom row, fed adult hermaphrodite treated with UV light to induce DNA damage. Asterisks, distal gonad ends. Arrowheads, example of an enlarged nucleus having elevated phospho-CDK-1. Similar results were observed using a different phospho-CDK-1 antibody (Calbiochem #219440, data not shown). (B) Time courses showing the mean number of M-phase cells per progenitor zone after food removal for hermaphrodites of the genotypes shown. Animals were starved from early adult. Grey curves represent seven replicates of wild type, reproduced from *Figure 2A*. Time zero indicates the start of food removal. Lower right corner, wildtype hermaphrodites exposed to 20 mM serotonin at the onset of food removal (0 hr) or 3 hr later. n = 46–452 gonadal arms per time point. Source data are available in *Figure 7—source data 1*.

The following source data and figure supplement are available for figure 7:

**Source data 1.** Starvation time courses of mutants and of wildtype hermaphrodites exposed to exogenous serotonin.

**Figure supplement 1.** Validation of phospho-specificity of anti-phospho-CDK-1.

phase of the cell cycle. Further, this quiescence maintains germline stem cells independent of GLP-1/Notch, a signal required for stem cell maintenance under conditions of active proliferation. Re-feeding causes germ cells to exit quiescence rapidly, and a requirement for GLP-1/Notch signaling similarly resumes.

## Cell-cycle quiescence promotes stem cell maintenance

Cell-cycle quiescence was once thought to be a near universal feature of stem cells in adult tissues (*Hall and Watt, 1989*; *Potten and Loeffler, 1990*). This view arose from the theory that biological systems ought to protect stem cells from the risks of DNA replication and led to the notion of quiescence as an inherent property of the stem cell fate. According to this model, tissues were maintained by a hierarchy of relatively quiescent master stem cells and their faster cycling but short-lived daughters. In recent years, however, work in several mammalian and invertebrate tissues has shown that quiescence is not a prerequisite for the stem cell fate (*Barker et al., 2010a*; *Crittenden et al., 2006*; *Doupe and Jones, 2013*; *Fuchs, 2009*; *Maciejowski et al., 2006*; *Simons and Clevers, 2011*). Some types of stem cells do not exhibit quiescence under conditions assayed (e.g. *Barker et al., 2010b*; *de Navascues et al., 2012*; *Snippert et al., 2010*), and still others vary their cell-cycle length in accordance with the physiological circumstances surrounding them (e.g. *Harrison and Lerner, 1991*; *Hartman et al., 2013*; *Lugert et al., 2010*; *Qiao et al., 2007*). Nevertheless, genetic or environmental perturbations that impact the cell cycle can also affect stemness (*Orford and Scadden, 2008*; *Pietras et al., 2011*; *Yilmaz et al., 2012*). Thus, a long-standing question in the field of stem cell biology has remained: Does cell-cycle quiescence play a role in maintaining the stem cell fate? Our results answer this question by showing that quiescence itself alters the genetic requirements for stemness: Actively dividing germline stem cells in *C. elegans* require GLP-1/Notch signaling for their maintenance (*Austin and Kimble, 1987*); we find that cell-cycle quiescence—induced by starvation, high NaCl, or absence of sperm—relieves this requirement (*Figure 6J*). Thus, cell-cycle quiescence maintains germline stem cells independent of the signal required for such maintenance under conditions of active proliferation.

The molecular mechanisms maintaining stem cells during periods of cell-cycle quiescence remain to be determined. Acting downstream of GLP-1/Notch signaling to maintain germline stem cells are the PUF-family translational repressors FBF-1 and FBF-2 (*Crittenden et al., 2002*) and the proteins of unknown molecular function LST-1 and SYGL-1 (*Kershner et al., 2014*). Quiescence might stabilize these stem cell regulators—for example, by inhibiting the protein degradation machinery linked to the cell-cycle. Alternatively, quiescence might substitute for the repressive effects of FBF-1 and FBF-2 by repressing translation on a global level. Global repression of translation is a conserved stress response (*Spriggs et al., 2010*), and the stress of starvation represses translation of at least a few genes in *C. elegans* (*Lascarez-Lagunas et al., 2014*). Another possibility is that stem cell maintenance might be regulated by a metabolite or metabolic process whose levels change during

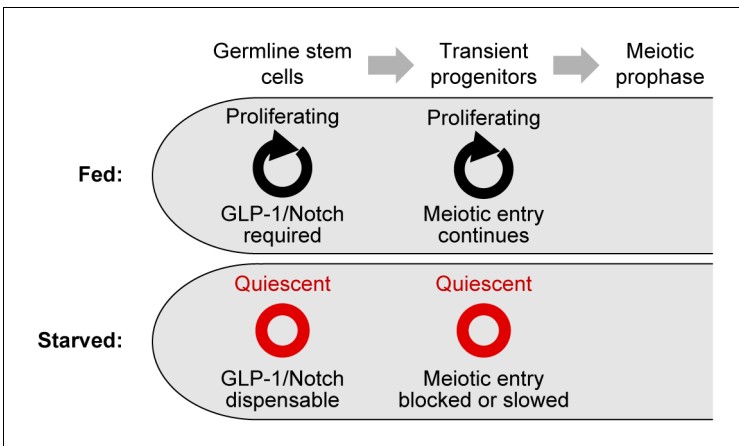

**Figure 8.** Summary of the effects of starvation on germ cell division, stem cell maintenance, and meiotic entry. Under fed conditions, mitotic cell divisions occur throughout the progenitor zone (*Crittenden et al., 2006*; *Maciejowski et al., 2006*); GLP-1/Notch is required for stem cell maintenance (*Austin and Kimble, 1987*); and transient progenitors enter the meiotic cell cycle if their movement out of the progenitor zone is restricted (*Cinquin et al., 2010*). Under starved conditions, germ cells become quiescent; GLP-1/Notch is dispensable for stem cell maintenance; and the meiotic entry of transient progenitors is slowed or blocked.

quiescence. Such connections between metabolism and developmental processes have been observed in a variety of vertebrate and invertebrate cell types (*Agathocleous and Harris, 2013*).

## Control of the G2-to-M transition by nutrients or growth factors may be a broadly conserved feature of the eukaryotic cell cycle

We find that adult germ cells in *C. elegans* do not arrest in G1 during starvation but instead progress slowly through S phase and arrest in G2. Most eukaryotic cells can transiently pause in G2 in response to DNA damage or microtubule disassembly (*Rieder, 2011*), but the G2-to-M transition has not been viewed as a point of cell-cycle control in response to growth factors or nutrients, largely because early studies in mammalian tissue culture showed that cell-cycle events in fibroblasts become independent of extracellular cues after entry into S phase (*Pardee, 1989*). Despite this view, the G2-to-M transition is emerging as the primary point of cell-cycle control in *C. elegans* germ cells, and the G2-to-M transition also responds to growth factors or nutrients in other systems.

*C. elegans* germ cells arrest in G2 during embryonic development (*Fukuyama et al., 2006*) and two larval diapause states—L1 diapause (*Fukuyama et al., 2006*) and dauer (*Narbonne and Roy, 2006*). Our results show that G2 arrest also occurs in starved adults (*Figure 3C*), and that the length of G2 can vary more than eightfold even under well-fed conditions (*Figure 3—figure supplement 1*). At the same time, G1 is very short (*Fox et al., 2011*, *Figure 3—figure supplement 1*), and cyclin E/cyclin-dependent kinase 2, which drives the G1-to-S transition (*Orford and Scadden, 2008*), is active throughout the cell cycle (*Fox et al., 2011*). These observations support a model of cell-cycle control in *C. elegans* germ cells in which regulation in response to extracellular cues relies on the G2-to-M transition.

G2 arrest in response to nutrient limitation has also been observed for fission yeast (*Costello et al., 1986*), budding yeast (*Laporte et al., 2011*), and *Tetrahymena* (*Cameron and Bols, 1975*). Moreover, the G2-to-M transition is a point of cell-cycle control in response to poor nutrient conditions in germline stem cells of *Drosophila* (*Hsu et al., 2008*; *LaFever et al., 2010*; *Roth et al., 2012*). Even under replete conditions, germline stem cells in *Drosophila* are thought to be paused in G2 (*Morris and Spradling, 2011*), as are a substantial fraction of *Drosophila* intestinal stem cells (*Zielke et al., 2016*). In mammals, cells can pause in G2 for prolonged periods of time before dividing in response to various stimuli (e.g. wounding, hormones) (reviewed in *Gelfant, 1977*). Such pausing in mammals has not been the subject of recent investigation, although the G2-to-M transition is known to be regulated by the growth factor IGF-1—for example, in mammalian uterine cells (*Adesanya et al., 1999*), oligodendrocyte progenitors (*Frederick and Wood, 2004*; *Min et al., 2012*), spermatogonial stem cells (*Wang et al., 2015*), and multiple myeloma cells (*Stromberg et al., 2006*). Proper timing of the G2-to-M transition is also essential during development (reviewed in *Bouldin and Kimelman, 2014*), with some populations of cells naturally held in G2 in both *Drosophila* and zebrafish (e.g. *Bouldin et al., 2014*; *Usai and Kimura, 1992*). Thus, the control of the G2-to-M transition by growth factors or nutrients may be a conserved feature of the eukaryotic cell cycle.

## *C. elegans* germline is a model for tissue plasticity and facultative stem cell quiescence

Tissues in adult organisms can be remarkably plastic in their ability to shrink and re-grow in response to changing physiological demands (e.g. *Bergtold, 1926*; *Secor and Diamond, 1998*). Such plasticity requires broad flexibility in a range of cellular behaviors, yet our understanding of tissue plasticity on a cellular level is limited, primarily because tractable models of tissue plasticity are only now being developed (e.g. *O'Brien et al., 2011*). The *C. elegans* germline presents such a model. This tissue undergoes dramatic shrinkage in adult hermaphrodites starved from the L4 larval stage, and such shrinkage is reversible upon re-feeding (*Angelo and Van Gilst, 2009*; *Seidel and Kimble, 2011*). Previous studies showed that germline shrinkage occurs in part through programmed cell death and oogenesis (*Angelo and Van Gilst, 2009*; *Seidel and Kimble, 2011*). This work establishes facultative stem cell quiescence as a third major force: Because germ cells stop dividing during starvation, cells lost to cell death and oogenesis are not replaced, thus causing the germline tissue to shrink. Quiescence also contributes to re-growth during re-feeding by ensuring that germ cells are able to re-enter the cell cycle rapidly in response to food. Rapid exit from quiescence is a

characteristic shared by analogous re-feeding responses in other animals, at least in the handful of examples where such responses have been examined at short time scales—for example, in the ovary of protein-limited *Drosophila* (*Hartman et al., 2013*; *Jouandin et al., 2014*), in the gut of fasted rats, squirrels, and chicks (*Aldewachi et al., 1975*; *Cameron and Cleffmann, 1964*; *Hagemann and Stragand, 1977*; *Kruman et al., 1988*), and in the retina of yolk-deprived frog embryos (*Love et al., 2014*). These observations stand in contrast to the comparatively longer times required for exit from quiescence in mammalian tissue culture (*Lum et al., 2005*; *Pardee, 1974*; *Soprano, 1994*; *Zetterberg and Larsson, 1985*) and suggest that in vivo models of quiescence may uncover new mechanisms of cell-cycle control.

## Materials and methods

### Strains

N2, CB4856 (*Hodgkin and Doniach, 1997*), TJ1 *cep-1(gk138) I* (*Consortium, 2012*), TG38 *aak-2 (gt33) X* (*Lee et al., 2008*), RB754 *aak-2(ok524) X* (*Narbonne and Roy, 2006*), JK5399 *aak-1 (tm1944) III; aak-2(ok524) X* (*Fukuyama et al., 2012*), MR507 *aak-2(rr48) X* (*Narbonne and Roy, 2006*), JK326 *par-4(it57ts) V* (*Watts et al., 2000*), GR1310 *akt-1(mg144gf) V* (*Paradis and Ruvkun, 1998*), CF1038 *daf-16(mu86) I* (*Lin et al., 1997*), NS3227 *daf-18(nr2037) IV* (*Mihaylova et al., 1999*), RB712 *daf-18(ok480) IV* (*Fukuyama et al., 2006*), JK5011 *daf-3(e1376) X* (*Patterson et al., 1997*), JK4971 *daf-5(e1386) II* (*da Graca et al., 2004*), IK130 *pkc-1(nj3) V* (*Sieburth et al., 2007*), CB169 *unc-31(e169) IV* (*Speese et al., 2007*), KP2018 *egl-21(n476) IV* (*Husson et al., 2007*), MT1241 *egl-21 (n611) IV* (*Husson et al., 2007*), VC461 *egl-3(gk238) V* (*Husson et al., 2006*), VC671 *egl-3(ok979) V* (*Husson et al., 2006*), JK4963 *bbs-5(q507) III* (*Lee et al., 2011*), JK4962 *bbs-5(gk537) III* (*Lee et al., 2011*), JK4960 *bbs-8(nx77) V* (*Blacque et al., 2004*), JK4964 *bbs-9(gk471) III* (*Chen et al., 2006*), JK4970 *tom-1(ok285) I* (*Gracheva et al., 2006*), CB1377 *daf-6(e1377) X* (*Perens and Shaham, 2005*), BS860 *unc-32(e189) glp-1(oz112gf)/dpy-19(e1259) glp-1(q172) III* (*Berry et al., 1997*), JK3182 *gld-3(q730) nos-3(q650)/mln1[mls14 dpy-10(e128)] II* (*Eckmann et al., 2004*), JK5098 *fog-1(q785) I/ hT2[qls48](I;III); glp-1(q224ts) III/hT2[qls48](I;III)* (*Morgan et al., 2010*), JK5336 *weSi2[Pmex-5::gfp:: his-58::tbb-2 3' UTR; Cbr-unc-119(+)] II; emb-30(tn377ts) III* (*Furuta et al., 2000*), JK4605 *glp-1 (q224ts) III* (*Kodoyianni et al., 1992*)

### Worm maintenance, synchronization, and staging

Unless otherwise noted, worms were maintained on nematode growth media spotted with *Escherichia coli* OP50 at 20°C. Nematode growth media contained 3 g/L NaCl, 2.5 g/L peptone, 20 g/L agar, 25 ml/L 1 M potassium phosphate buffer (1 M $K_2HPO_4$ mixed with 1 M $KH_2PO_4$ to reach a pH of 6.0), 1 mM $CaCl_2$, 1 mM $MgSO_4$, 5 µg/ml cholesterol, and 2 µg/ml uracil. Worms were synchronized by bleaching gravid hermaphrodites for 5–8 min in a 1:2:12 solution of 5 M NaOH: household bleach: M9 (3 g/L $KH_2PO_4$, 6 g/L $NaHPO_4$, 5 g/L NaCl, and 1 mM $MgSO_4$). Embryos were allowed to hatch overnight in M9 in an aerated flask, with shaking at ~170 rpm. L1 larvae were then plated onto 10-cm plates, at a density of ~1000 per plate, and grown to the appropriate developmental stage.

We staged animals as 'early adult' when hermaphrodites had molted into adulthood and recently begun to ovulate, with most hermaphrodites containing one or two embryos in utero, but with some hermaphrodites containing zero embryos or up to four embryos. Animals with germline tumors (and therefore no ovulation) were staged according to the ovulation status of their non-tumorous siblings. At 20°C, early adulthood was reached ~48–52 hr after L1 feeding. For staging of L4 animals, we examined the extent to which the hermaphrodite gonad had migrated from the loop towards the vulva. We define populations as 'early L4' when the gonad had migrated ~1/4 of this distance and 'mid-L4' when the gonad had migrated ~1/2–3/4 of this distance. Males were staged according their hermaphrodite siblings.

### Food removal and re-feeding

Food was removed by gently washing animals from plates with M9, pelleting animals by spinning at 100–200g for ~1 min, and then washing animals 3–6 additional times with M9, using 15 ml M9 per wash per 1000–3000 animals. Animals were then deposited onto unseeded 10-cm plates and gently spread across plates such that all liquid was absorbed into the plate within 5 min. Media for

starvation plates contained 3 g/L NaCl, 25 g/L agar, 25 ml/L 1 M potassium phosphate buffer (see above), 1 mM CACL$_2$, 1 mM MgSO$_4$, and 5 µg/ml cholesterol. This starvation procedure lasted ~10–15 min from start to finish, with time zero being the moment at which M9 was first added to the bacterially seeded plates. Mock food removal was performed in the same manner, except that animals were washed in M9 + ~0.5% OP50 and deposited onto bacterially seeded plates. In most experiments, animals were starved at densities of 500–1000 animals per 10-cm plate. Exceptions were starvations beginning from L4, in which animals were starved at densities of 2000–3000 per 10-cm plate, and experiments requiring that animals be hand-picked, in which animals were starved at densities of <500 per 10-cm plate.

Re-feeding was performed by washing animals from starvation plates with M9 + ~0.5% OP50, spinning at 100–200g for ~1 min to pellet animals, and then depositing animals onto 10-cm nematode growth media plates seeded with OP50. Animals were spread across the plate such that all liquid was absorbed into the plate within 5 min.

## Genotypes requiring non-standard growth conditions

*glp-1(q224ts)* and *fog-1(q785); glp-1(q224ts)*: Animals were grown at 15°C. All plates and M9 solutions used to handle animals were pre-equilibrated to 15°C. Animals were synchronized as described above, but the bleaching protocol was modified, as follows, because *glp-1(q224)* embryos are bleach-sensitive. Gravid hermaphrodites were incubated in bleaching solution for ~1 min to kill the adult hermaphrodites but allow their carcasses to remain intact. Embryo-containing carcasses were incubated at 15°C for 4–8 hr. Carcasses were then bleached again, for 3–4 min, to liberate embryos. Hatching of L1s in M9 and was performed at 15°C and extended to 36–40 hr, to account for longer embryonic development times at 15°C. At 15°C, animals reached the early adult stage ~90–96 hr after L1 feeding.

*par-4(it57ts)*: Animals were grown at 15°C until the L3 stage, and then shifted to 25°C until animals reached early adult. The *par-4(it57ts)* starvation time course was performed at 25°C.

*aak-1(tm1944); aak-2(ok524)*: Animals were grown at 15°C until the early L4 stage, and then transferred to 20°C until animals reached early adult. Growth at 15°C was used because *aak-1(tm1944); aak-2(ok524)* animals grown at 20°C showed high sterility. Additionally, an alternate synchronization protocol was used because hatching of *aak-1(tm1944); aak-2(ok524)* L1s in M9 causes sterility (*Fukuyama et al., 2012*). Early adults were bleached according to the protocol described for *glp-1 (q224ts)*, and embryos were deposited directly onto food.

*emb-30(tn377ts)*: Animals were grown at 15°C prior to temperature shifts.

## Antibody and DAPI staining

Gonads were dissected in M9 + 0.1% Tween-20 + 0.25 mM levamisole. For GLD-1 and phospho-histone H3 staining, gonads were fixed in PBS + 3% paraformaldehyde + 0.1% Tween-20 (PBSTween) for 30 min, followed by −20°C methanol for 15 min (anti-GLD-1 and anti-HIM-3) or ≥15 min (anti-phospho-histone H3). For phospho-CDK-1 staining, gonads were fixed in PBSTween + 3.7% formaldehyde for 10 min, followed by −20°C methanol for 5 min. Gonads were blocked for 30 min at room temperature in PBSTween + 3–5% normal donkey serum (anti-GLD-1 and anti-phospho-CDK-1) or 1–3% bovine serum albumin (anti-HIM-3 and phospho-histone H3). Incubations with primary antibodies were performed overnight at 4°C, with antibodies diluted in blocking solution. Dilutions were as follows: mouse anti-phospho-histone H3 (Cell Signaling Technology, Danvers, MA, #9706), 1/150; rabbit anti-GLD-1 (*Cinquin et al., 2010*), 1/100; rabbit anti-HIM-3 (Novus Biologicals, Littleton, CO, #53470002), 1/200; rabbit anti-phospho-CDK-1 Thr14/Tyr15 (Santa Cruz, Dallas, Texas, #sc-28435-R) (*Rahman et al., 2014*), 1/200; rabbit anti-phospho-CDK-1 Tyr15 (Calbiochem, San Diego, CA, #219440) (*Hachet et al., 2007*), 1/100. Incubations with secondary antibodies were performed for 1–2 hr at room temperature, using Cy-3 donkey anti-mouse (Jackson ImmunoResearch, Westgrove, PA, #715-165-151) or Cy-3 donkey anti-rabbit (Jackson ImmunoResearch #711-165-152), diluted 1/1000. Gonads were mounted in Vectashield containing DAPI (Vector Labs, Burlingame, CA, #H-1200). For DAPI staining in the absence of antibody staining, gonads were fixed as for phospho-histone H3 staining, and then mounted in Vectashield containing DAPI.

To test the phospho-specificity of anti-phospho-CDK-1, gonads were dissected and fixed, as above. After fixation, gonads were treated with 20 U/µl Lambda protein phosphatase (NEB, Ipswich,

MA, #P0753S) in protein metallophosphatase buffer (50 mM HEPES, pH 7.5, 100 mM NaCl, 2 mM DTT, 0.01% Brij 35, and 1 mM $MnCl_2$) for 1 hr at 30°C Control gonads were treated the same, but Lambda protein phosphatase was omitted from the reaction. Gonads were blocked and stained with anti-phospho-CDK-1 as above.

## Imaging

Unless otherwise noted, images were obtained on a Leica SP8. In all experiments, identical imaging conditions and brightness adjustments were used across samples.

## Counting M-phase cells

Unless otherwise noted, M-phase cells were scored by examining gonads for phospho-histone H3[+] cells at 63× magnification. In all experiments, a subset of gonads was examined via DAPI staining to confirm the correspondence between phospho-histone H3[+] cells and mitotic figures. In germline tumors, M-phase cells occurring in the distal-most 20 rows of germ cells (i.e. the region corresponding to the progenitor zone in wildtype gonads) were excluded from analysis. Additionally, we excluded germline tumors with patches of differentiation (as assessed by DAPI staining), which sometimes occurred in *glp-1(oz112gf)* animals.

## EdU labeling

EdU labeling was performed by soaking or by feeding. Soaking was used for starvation time courses (*Figure 3A*) and for testing whether G1 versus G2 cells could be distinguished by nuclear size (*Figure 3—figure supplement 2*). Feeding was used for measuring cell-cycle length in fed animals (*Figure 3—figure supplement 1*). For the soaking procedure, animals were incubated with rocking in M9 + 0.1% Tween-20 + 1 mM EdU for 15 min at room temperature. Gonads were dissected as for antibody staining and fixed in 3% paraformaldehyde in PBSTween for 30 min, followed by −20°C methanol for ≥15 min. Gonads were blocked in PBSTween + 3% bovine serum albumin for 30 min at room temperature. Click-iT reactions were performed using the Click-iT EdU Alexa Fluor 488 Imaging Kit (Invitrogen, Carlsbad, CA, #C10337), according to the manufacturer's instructions, except that two back-to-back half reactions of 250 μl volume were performed. Gonads were mounted in Vectashield containing DAPI.

EdU labeling by feeding was performed similar to previous studies (*Crittenden et al., 2006*; *Fox et al., 2011*; *Morgan et al., 2010*). *E. coli* strain MG1693 was grown overnight at 37°C in M9 minimal media (3 g/L $KH_2PO_4$, 6 g/L $Na_2HPO_4$, 0.5 g/L NaCl, 1 g/L $NH_4Cl$, 2 mM $MgSO_4$, 0.1 mM $CaCl_2$, 0.4% glucose, and 1 μg/ml thiamin) supplemented with 5 μg/ml thymine. This culture was diluted 1:50 in M9 minimal media supplemented with 0.5 μM thymidine and 20 μM EdU and grown for 32 hr at 37°C. Cells were re-suspended in ~1/100th of their original volume in M9, and then spread onto 6-cm plates, using 100 μl of *E. coli* solution per plate. Plate media was identical to standard nematode growth media except that 60 μg/ml carbenicillin was added, peptone was omitted, and agar was exchanged for 12 g/L agar + 6 g/L agarose. Plates were seeded 1 day prior to adding worms. Worms were transferred to plates for the required period of time, and then gonads were dissected and processed as above.

## Propidium iodide staining and its quantification

EdU-labeled gonads were stained with propidium iodide to measure DNA content and to allow for simultaneous imaging of Alexa Fluor 488 and DNA. Propidium iodide staining was performed by adding two steps to the aforementioned EdU-labeling protocol. First, prior to the blocking step, gonads were incubated in PBSTween + 20 μg/ml RNase A for 1 hr at 37°C. Second, after the Click-iT reaction, gonads were incubated for 30 min at room temperature in PBSTween + 50 μg/ml propidium iodide.

To quantify propidium iodide staining, gonads were imaged at 63× magnification on a Zeiss LSM510 laser-scanning confocal microscope, with a z-stack interval of 0.37–0.39 μm. Pixel intensity per nucleus was calculated as the summation of all pixel intensities within a best-fit cylinder whose height matched the height of the focal nucleus in the z-dimension and whose cross-sectional diameter matched the largest dimension of the focal nucleus in the x–y dimension. Cylinders were defined manually by drawing circles around nuclei in ImageJ. This quantification method is undeniably crude

because cylinders often included portions of neighboring nuclei. Nevertheless, this method allowed us to distinguish non-S-phase cells as having a DNA content less than S-phase cells (G1) or a DNA content greater than S-phase cells (G2) (*Figure 3—figure supplement 2*).

## Use of nuclear size to distinguish G1 versus G2 cells

Cells were classified as G1, S-phase, or G2 by a combination of EdU labeling (to mark S-phase cells) and nuclear size. This method has not been used previously—although others have noted a correlation between cell-cycle stage and nuclei size (*Chiang et al., 2015*; *Fukuyama et al., 2006*; *Lawrence et al., 2015*)—and we justify its use here. In pilot experiments involving EdU labeling and DNA quantification, we noticed a correlation between cell-cycle stage and nuclear size: G1 and early S-phase nuclei were small, G2 and late S-phase nuclei were large, and mid S-phase nuclei were intermediate in size (*Figure 3—figure supplement 2*). Additionally, G1 nuclei nearly always occurred in pairs, consistent with G1 being very short (*Fox et al., 2011*). The size difference between G1 and G2 nuclei was large enough that in EdU-labeled gonads, G1 and G2 nuclei could be distinguished by eye (n > 100 nuclei, from a total of seven progenitor zones). To test the accuracy of this method more thoroughly, we obtained z-stack images of EdU-labeled, propidium iodide-stained progenitor zones from three fed early adult hermaphrodites and three hermaphrodites starved from early adult for 3.5 hr. We first classified each cell as S-phase (EdU$^+$), G1 (EdU$^-$ and having a nuclear size equal to or smaller than the smallest EdU$^+$ cells), or G2 (EdU$^-$ and having a nuclear size equal to or larger than the largest EdU$^+$ cells). We then quantified propidium iodide staining (a measure of DNA content) and compared our 'by size' classification to the classification given by propidium iodide staining (*Figure 3—figure supplement 2*). For all cells in all six progenitor zones, the two classification systems matched perfectly (*Figure 3—figure supplement 2*). We therefore used the 'by size' classification system for counting G1, S-phase, and G2 cells throughout.

## Counting G1, S-phase, and G2 cells

EdU-labeled progenitor zones were imaged at 63× magnification with a z-stack interval of 1 μm. Cells were counted as belonging to the progenitor zone if they had an interphase or M-phase chromosome morphology and if their midpoint was located distal to a cross-sectional line drawn through the midpoint of the second most distal 'crescent' cell (i.e. the second most distal meiotic prophase cell). Progenitor-zone cells were classified as G1 (EdU$^-$ and nuclear size equal to or smaller than the smallest EdU$^+$ cells), S-phase (EdU$^+$), G2 (EdU$^-$ and nuclear size equal to or larger than the largest EdU$^+$ cells), or M-phase (mitotic figures). Classifications were recorded using the Cell Counter plug-in for ImageJ (http://rsb.info.nih.gov/ij/plugins/cell-counter.html), by marking each cell in all z-slices in which it was observed. A custom R script was then used to identify marks belonging to the same cell.

## Measuring cell-cycle parameters in fed animals

Cell-cycle parameters in fed animals were determined by first measuring the length of G2. G2 was measured by labeling animals with EdU (via feeding) and calculating the fraction of M-phase cells (mitotic figures) that were EdU$^+$ over time (equation 1). Next, the length of G2 was combined with the G2 index to calculate the total length of the cell cycle (equation 2). The lengths of G1, S-phase, and M-phase were calculated by multiplying the total length of the cell cycle by the G1, S-phase, or M-phase indices (equations 3–5). Calculations of the maximum total length of the cell cycle depend on assumptions about covariance between the length of G2 and the length of M + G1 + S. If cells having a longer G2 are assumed to have a proportionally longer M + G1 + S, then the maximum length of the cell cycle for 95% or 100% of cells is given by equation 6. If cells having a longer G2 are not assumed to have a proportionally longer M + G1 + S, then the maximum length of the cell cycle for 95% or 100% of cells is given by equation 7. We performed both calculations.

1. Median length of G2 = Time at which 50% of M-phase cells were EdU$^+$.
2. Median total length of cell cycle = Median length of G2/G2 index.
3. Median length of G1 = Median total length of cell cycle * G1 index.
4. Median length of S-phase = Median total length of cell cycle * S-phase index.
5. Median length of M-phase = Median total length of cell cycle * M-phase index.

6. Maximum total length of cell cycle for 95% or 100% of cells = Time at which 95% or 100% of M-phase cells were EdU$^+$/G2 index.
7. Maximum total length of cell cycle for 95% or 100% of cells = Median total length of cell-cycle * (M + G1 + S index) + time at which 95% or 100% of M-phase cells were EdU$^+$.

Cell-cycle length in fed animals was measured in early adult hermaphrodites and in adult hermaphrodites aged 24 hr post mid-L4. For calculations involving early adults, indices for each cell-cycle phase were derived from the 0-hr time point of the time course in *Figure 3*. For calculations involving animals aged 24 hr post mid-L4, indices were derived from the 0.5-hr time point of the time course in *Figure 3—figure supplement 1*, for consistency with *Fox et al. (2011)*.

## Measuring distance from the gonad distal tip to the first 'crescent' germ cell

Gonads were stained with DAPI and examined at 63× magnification. In a central focal plane, one edge of the gonad was chosen at random, and the distal-most 'crescent' germ cell along that edge was identified. The number of germ cells (along the gonad edge) between this first 'crescent' cell and the distal tip of the gonad was counted.

## UV treatment

L4 hermaphrodites close to the adult molt were transferred to an unseeded plate and exposed to 100 J/m$^2$ of 254 nm UV light in Spectrolinker XL-1000 UV Crosslinker. Animals were immediately returned to food and incubated for 8 hr at 20°C before dissection. Animals were adults at the time of dissection.

## Serotonin exposure

Serotonin creatinine sulfate was dissolved in M9 to a concentration of 50 mg/ml. This solution was spread onto starvation plates to a final concentration of 20 mM serotonin. Plates were incubated for at least 1 hr before worms were added. To expose animals to serotonin at the onset of starvation, animals were deposited directly onto serotonin plates after food removal. To expose animals to serotonin after 3 hr of starvation, animals were starved for 3 hr on standard starvation plates and then were transferred to serotonin plates via washing in M9 + 0.01% Tween-20.

## cdk-1 RNAi

RNAi was performed by feeding. A *cdk-1* RNAi clone and the empty RNAi vector (L440) were obtained from the Ahringer library (*Kamath and Ahringer, 2003*) and grown overnight at 37°C in liquid Luria Broth + 60 µg/ml carbenicillin + 10 µg/ml tetracycline. Cells were concentrated fivefold, and then spotted onto plates containing nematode growth media supplemented with 60 µg/ml carbenicillin, 10 µg/ml tetracycline, and 1 mM IPTG. Plates were spotted 1 day before adding worms.

## Temperature shifts

Temperature shifts were performed by transferring plates of worms from 15°C to 25°C or the reverse. To expedite equilibration of plates to the new temperature, plates were buried in a single layer within a 5 × 7 × 13 inch box full of unseeded plates pre-equilibrated at the new temperature.

For temperature shifts of fed *glp-1(q224ts)* animals, fed early adult hermaphrodites were shifted to 25°C. Gonads were dissected every 30–60 min after the shift (for *Figure 6H*) or after 8 hr (for *Figure 6A, I*). Alternately, animals were maintained at 25°C for 8 hr, then returned to 15°C for 2–3 days (*Figure 6A*).

For temperature shifts of *cdk-1* RNAi-treated animals (*Figure 6B* and *Figure 6—figure supplement 1*), *glp-1(q224ts)* animals were grown at 15°C (on OP50) to mid L4, and then transferred to *cdk-1* RNAi plates or L440 plates for 42 hr hours. Plates were then shifted to 25°C for 8 hr, and then returned to 15°C for 2–3 days. At the beginning of the temperature shift, animals were adults, and germ cells in the progenitor zone had uniformly arrested in interphase, as evidenced by the absence of mitotic figures. Some nuclei in the progenitor zone had slightly enlarged, characteristic of *cdk-1* RNAi-induced cell-cycle arrest (*Jeong et al., 2011*). Nuclear morphology was otherwise normal (*Figure 6—figure supplement 1*).

For temperature shifts of starved *glp-1(q224ts)* animals, hermaphrodites were starved at 15°C from early adult for 2–4 hr or from mid L4 for 24 hr. For *Figure 6C–D*, starved animals were shifted to 25°C for 8, 16, or 24 hr, returned to 15°C for 8–12 hr, then re-fed at 15°C for 2–3 days. At the beginning of the temperature shift, animals starved from L4 were adults. For *Figure 6G–H*, animals were shifted to 25°C for 8 hr, and then re-fed at 25°C. For *Figure 6I*, animals were shifted to 25°C for 8 hr.

For temperature shifts of *glp-1(q224ts)* animals exposed to high NaCl (*Figure 6E*), early adult hermaphrodites were transferred to plates containing high NaCl media and incubated at 15°C for 2 hr. Animals were then shifted to 25°C for 8 hr, returned to 15°C for 12 hr, and then transferred to plates containing standard media for 2–3 days. High NaCl media were identical to standard media, except that it contained a total of 17.4 g/L (300 mM) NaCl.

For temperature shifts of (unmated) *fog-1(q785);glp-1(q224ts)* animals (*Figure 6F*), hermaphrodites aged 40 hr post mid L4 were shifted to 25°C for 8 hr, and then returned to 15°C for 2–3 days. Animals aged 40-hr post mid L4 were used because the reduced mitotic index caused by the absence of sperm is not fully evident at the early adult stage (40-hr post mid-L4 at 15°C is roughly equivalent to 24-hr post mid-L4 at 20°C). For temperature shifts of mated *fog-1(q785);glp-1(q224ts)* animals, *fog-1(q785);glp-1(q224ts)* mid L4 hermaphrodites were transferred to plates with CB4856 males at a ratio of 1:2. After 40 hr of mating, hermaphrodites wearing copulatory plugs were transferred to fresh plates at 15°C, and then shifted to 25°C for 8 hr.

Temperature shifts of *emb-30(tn377ts)* animals are described below.

## *emb-30(tn377ts)* experiments and analyses

*emb-30*(*tn377ts*) animals were grown at 15°C to mid L4, and then starved at 15°C for 24 hr, by which time animals were adults. Animals were then shifted to 25°C for 14 hr and either re-fed or maintained in starvation. Gonads were dissected immediately before re-feeding and at 1-hr intervals 2–6 hr after re-feeding. Gonads were also dissected after 6 hr of continued starvation. Gonads were stained for phospho-histone H3 and GLD-1 and imaged at 63× magnification with a z-stack interval of 1 μm. Cells were identified using IRISES (*Vogel et al., 2014*), followed by manual correction using the Cell Counter plug-in in Image J. Cells were classified as interphase, M-phase, or 'crescent' (i.e. meiotic prophase) according to chromosome morphology and phospho-histone H3 staining. Occasionally, at the later re-feeding time points, cells were observed bypassing the metaphase arrest; for the purposes of this experiment, such cells were classified as M-phase. Cells distal to the 'crescent'/'non-crescent' boundary were defined as cells whose midpoints were located distal to a cross-sectional line drawn through the midpoint of the second most distal 'crescent' cell. Relative positions of cells along the distal-to-proximal axis were determined by collapsing cell positions along the z-axis and fitting these positions to a second-degree polynomial curve. Positions along this polynomial curve closest to each germ cell and closest to the distal tip cell were identified by solving, for each cell, the polynomial whose roots minimize this distance. Using this new set of positions, distances between the distal tip cell and each germ cell were calculated. Cells were then ranked according to these distances.

### Sample sizes

In experiments requiring image acquisition, an attempt was made to examine at least 20 gonads. In other experiments, an attempt was made to examine at least 50 gonads.

### Plots

Plots were generated in part using the ggplot package for R (ggplot2.org).

## Acknowledgements

We thank Jadwiga Forster for media preparation, Peggy Kroll-Conner for worm maintenance, and Sarah Crittenden, Kim Haupt, Aaron Kershner, and Erika Sorensen for comments on the manuscript. Some strains were provided by the *Caenorhabditis* Genetics Center, which is funded by National Institutes of Health Office of Research Infrastructure Programs (P40 OD010440). JK is an investigator of the Howard Hughes Medical Institute. HSS was supported by an Ellison Medical Foundation Fellowship of the Life Science Research Foundation.

# Additional information

## Funding

| Funder | Grant reference number | Author |
|---|---|---|
| Howard Hughes Medical Institute | | Judith Kimble |
| Ellison Medical Foundation | Life Science Research Foundation Fellowship | Hannah S Seidel |

The funders had no role in study design, data collection and interpretation, or the decision to submit the work for publication.

## Author contributions

HSS, Conception and design, Acquisition of data, Analysis and interpretation of data, Drafting or revising the article; JK, Conception and design, Analysis and interpretation of data, Drafting or revising the article

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
