## [Decision Letter]

Thank you for submitting your work entitled "Cell-cycle quiescence maintains germline stem cells independent of GLP-1/Notch signaling in *Caenorhabditis elegans*" for peer review at *eLife*. Your submission has been favorably evaluated by Sean Morrison (Senior editor), Alejandro Sánchez Alvarado (Reviewing editor), and two reviewers, one of whom, David Greenstein, has agreed to reveal his identity.

The reviewers have discussed the reviews with one another and the Reviewing editor has drafted this decision to help you prepare a revised submission.

Summary:

In this beautifully and clearly written manuscript, Seidel and Kimbleaim to better understand the molecular mechanisms by which nutrition affects stem cells, a critically important but incompletely understood process in animal biology. As a paradigm, the authors studied the effect of starvation on the adult germline stem cells of *C. elegans*, and report that starvation blocks stem cells in G2, and that this block is reversible upon re-feeding. The authors provide experimental evidence indicating that GLP-1/Notch signaling is not needed under conditions of quiescence, and that the starvation response of adult germline stem cells is independent of known signaling pathways shown to mediate nutritional sensing, such as insulin signaling, TGF-β signaling, and neuropeptide signaling. Based on these and other data Seidel put forward a model of starvation regulation they refer to as "facultative quiescence" defined by a slow S phase that is coupled to a block in M phase, and which is independent of GLP-1/Notch Signaling. Overall, the data and methods introduced in this article not only expand the methodological toolkit to study germline stem cells, but also contribute in significant ways to our understanding of the nutritional regulation of animal stem cells.

Essential revisions:

The experiments are very carefully done and clearly presented. This work lays an important foundation for future molecular genetic studies to address mechanism. The manuscript has a few shortcomings, however, which can for the most part be addressed textually. The manuscript should do a better job of incorporating two recent advances in this field. The first, from one of the reviewers, Dr. Greenstein (Starich et al., 2014) in which it was shown that soma germline gap junctions are as important as the GLP-1/Notch pathway for controlling germline stem cells. In the absence of these junctions, germ cells can neither proliferate nor differentiate, even if the GLP-1/Notch pathway is constitutively activated. These gap junctions enable the transit of small molecules, ions, and metabolites between germline and soma, thereby providing a potential mechanism for nutritional inputs to the germline. The second is a major paper from Tim Schedl's lab (Fox and Schedl, 2015), which substantially refines our understanding of germline stem cell behavior in *C. elegans*. After this publication, it seems questionable (or controversial at the very least) whether it is correct to refer to a proximal population of "transit-amplifying cells" in this system (as in Figure 1 here). A more complete consideration of Fox and Schedl (2015) raises an alternative interpretation of the authors' glp-1/Notch temperature-shift experiment (Major Point 1, below). The authors should consider the following specific points:

1) The authors interpret the result that starvation in *glp-1(ts)* does not cause cells to enter meiosis at the non-permissive temperature as meaning that "starved animals maintain germline stem cells independent of GLP-1/Notch." Given the results of Fox and Schedl (2015) that the majority of *glp-1(ts)* germ cells complete a single division before entering meiosis, it seems quite possible this is a consequence of the G2 arrest of cells after starvation. Upon moving back to 15C and re-feeding, germ cells complete their mitotic division and return to a situation in which GLP-1/Notch signaling is re-established. The authors interpret a *cdk-1*(RNAi) experiment as evidence that a G2 block is insufficient to prevent meiotic entry. Inspection of the Methods section associated with this experiment (subsection “Temperature shifts”, third paragraph) does not provide sufficient information as to the reliability of the method to score meiotic entry in these gonads by DNA morphology. The authors should consider evaluating meiotic entry by analyzing the recruitment of meiosis-specific chromosomal proteins to chromatin (e.g., HIM-3, etc.). At the very least, readers should be able to see for themselves what these gonads look like. The authors seem to be claiming that GLP-1/Notch signaling becomes non-essential during starvation conditions, not that it is turned off. If the latter was the case, then reporters for *sygl-1* and *lst-1* might be revealing.

2) The Introduction needs some rewriting. For example, readers would benefit from a more thorough review of the *C. elegans* literature and that of *Drosophila*, as pertains to how nutrition affects germ cells.

3) In the Introduction, or in the Results section (near the end of the subsection “During starvation, germ cells progress slowly through S phase and arrest in G2”), point out that PGCs arrest in G2 during embryogenesis and remain arrested in G2 when the newly hatched larvae are starved, only undergoing proliferation upon feeding (Fukuyama et al., 2006).

4) Figure 7 and the subsection “Starvation-induced quiescence is distinct from cell-cycle arrest induced by DNA-damage”: that the "phosphor-inhibited CDK-1" antibody reports only P-CDK-1 and does not cross-react with unphosphorylated CDK-1 in the immunostaining experiment has not been validated here or in Craig et al. (2012). The authors should present a validation of this reagent.

---

## [Author Response]

Essential revisions:

*The experiments are very carefully done and clearly presented. This work lays an important foundation for future molecular genetic studies to address mechanism. The manuscript has a few shortcomings, however, which can for the most part be addressed textually. The manuscript should do a better job of incorporating two recent advances in this field. The first, from one of the reviewers, Dr. Greenstein (Starich et al., 2014) in which it was shown that soma germline gap junctions are as important as the GLP-1/Notch pathway for controlling germline stem cells. In the absence of these junctions, germ cells can neither proliferate nor differentiate, even if the GLP-1/Notch pathway is constitutively activated. These gap junctions enable the transit of small molecules, ions, and metabolites between germline and soma, thereby providing a potential mechanism for nutritional inputs to the germline.*

Good suggestion. We have included this information in our expanded Introduction.

*The second is a major paper from Tim Schedl's lab (Fox and Schedl, 2015), which substantially refines our understanding of germline stem cell behavior in* C. elegans. *After, this publication, it seems questionable (or controversial at the very least) whether it is correct to refer to a proximal population of "transit-amplifying cells" in this system (as in Figure 1 here). A more complete consideration of Fox and Schedl (2015) raises an alternative interpretation of the authors' glp-1/Notch temperature-shift experiment (Major Point 1, below).*

Fox and Schedl (2015) present a careful and detailed analysis of germline stem cell behavior after loss of GLP-1/Notch. They report that when GLP-1/Notch signaling is removed, germ cells undergo one (or sometimes two) divisions before entering meiotic prophase. This finding is fully consistent with our previous understanding of cells in the proximal progenitor zone being in a transitional yet mitotically dividing state, and this finding is also consistent with the results of our current manuscript.

The observations of Fox and Schedl (2015) lead them to hypothesize that germ cells in mitotic S-phase or G2 are not able to enter meiotic prophase without passing through a cell division. Their hypothesis raised an alternative interpretation of our *glp-1*/Notch temperature-shift experiments, but their hypothesis was directly refuted by our *cdk-1* RNAi experiments. The reviewers raise questions about our *cdk-1* RNAi experiments and ask for further confirmation of these experiments using HIM-3 staining (Major Point 1, below). We performed HIM-3 staining, as requested, and found that HIM-3 staining supported our original conclusions. Thus, our results do not support the interpretation suggested by Fox and Schedl (2015).

The reviewer also uses Fox and Schedl (2015) to question our use of the term ‘transit-amplifying cells’ to refer to cells in the proximal progenitor zone. In response to this criticism, we sought a suitable replacement term and selected ‘transient progenitors’ as a possibility. This term follows the suggestion of Fox and Schedl (2015) to use ‘progenitors’, with the addition of ‘transient’ to convey the idea of a transitional state.

In selecting a term to refer to germ cells in the proximal progenitor zone, we sought to convey two ideas, neither of which is controversial. First, we wished to convey the idea that cells in the proximal progenitor zone are in the mitotic cell cycle. This idea is supported by many studies, including Fox and Schedl (2015). Second, we wished to convey the idea that cells in the proximal progenitor zone are in a transitional state. This idea is supported by three key observations: (i) cells in the proximal progenitor zone have higher levels of the meiosis-associated protein GLD-1 than cells in the distal progenitor zone (many studies); (ii) cells in the proximal progenitor zone have lower levels of the stem cell regulators *sygl-1* and *lst-1* than cells in the distal progenitor zone (Kershner et al., 2014); and (iii) cells in the proximal progenitor zone enter the meiotic cell cycle when their movement out of the progenitor zone is restricted, whereas cells in the distal progenitor zone arrest in the mitotic cell cycle and do not enter meiosis with that treatment (Cinquin et al., 2010). Based on these multiple lines of evidence, we sought terminology that would convey the idea that cells in the proximal progenitor zone are in a transitional yet mitotically dividing state. The term ‘transit-amplifying cells’ terms has long been used in the stem cell literature to mean just that – progeny of stem cells that continue to divide mitotically but that have limited division potential and have begun to differentiate.

We have now replaced ‘transit-amplifying cells’ with ‘transient progenitors’ throughout the revised manuscript. Yet we have mixed feelings about this replacement. The reviewers argue that ‘amplifying’ may mislead readers about the number of divisions occurring in the proximal progenitor zone. This problem was addressed in our original manuscript by informing readers that the number of divisions in this zone is one or two (and this information is retained in our revised manuscript). Moreover, the term ‘transit-amplifying cells’ has been used throughout the vertebrate literature to refer to cells with similar properties to cells in the proximal progenitor zone. Does the confusion caused by ‘amplifying’ outweigh the confusion caused by adding a new term to the literature? We ask the reviewers to consider the pros and cons of the change in terminology and the need for a new term. Our preference is to use the term ‘transit-amplifying cells’, but we agree to replace it with ‘transient progenitors’ if the reviewers remain unconvinced.

*The authors should consider the following specific points:*

*1) The authors interpret the result that starvation in* glp-1(ts) *does not cause cells to enter meiosis at the non-permissive temperature as meaning that "starved animals maintain germline stem cells independent of GLP-1/Notch." Given the results of Fox and Schedl (2015) that the majority of* glp-1(ts) *germ cells complete a single division before entering meiosis, it seems quite possible this is a consequence of the G2 arrest of cells after starvation. Upon moving back to 15C and re-feeding, germ cells complete their mitotic division and return to a situation in which GLP-1/Notch signaling is re-established. The authors interpret a* cdk-1*(RNAi) experiment as evidence that a G2 block is insufficient to prevent meiotic entry. Inspection of the Methods section associated with this experiment (subsection “Temperature shifts”, third paragraph) does not provide sufficient information as to the reliability of the method to score meiotic entry in these gonads by DNA morphology. The authors should consider evaluating meiotic entry by analyzing the recruitment of meiosis-specific chromosomal proteins to chromatin (e.g., HIM-3, etc.). At the very least, readers should be able to see for themselves what these gonads look like. The authors seem to be claiming that GLP-1/Notch signaling becomes non-essential during starvation conditions, not that it is turned off. If the latter was the case, then reporters for* sygl-1 *and* lst-1 *might be revealing.*

As requested, we have now performed HIM-3 staining in *glp-1(q224ts)* animals treated with *cdk-1* RNAi and shifted to the restrictive temperature. We observed that HIM-3 was recruited to chromosomes following the temperature shift (Figure 5–figure supplement 1). Thus, cell-cycle arrest is insufficient to inhibit meiotic entry. This result strengthens our conclusion that the ability of quiescence to maintain germline stem cells in absence of GLP-1/Notch is not simply an effect of cell-cycle arrest.

We also note that Cinquin et al. (2010) reported entry of cell cycle-arrested germ cells into meiotic prophase without DNA replication and without mitotic division:

“In unablated controls, the distal germ cells did not enter meiosis, as assayed by absence of abundant GLD-1 (100%, n = 4; Figure 2), absence of crescent-shaped DNA (100%, n = 10; Figure 2), and absence of chromosomal HIM-3 (100%, n = 6). By contrast, when the DTC was ablated, the distal germ cells accumulated abundant GLD-1 (100%, n=5; Figure 2) and entered meiotic prophase, as assayed by both crescent-shaped DNA (100%, n = 9) and HIM-3 localization (100%, n = 4; Figure 2). We do not understand the mechanism by which germ cells escape cell cycle arrest after DTC ablation, but note that some nuclei with crescent-shaped DNA and chromosomal HIM-3 had not undergoneDNAsynthesis after DTCablation (see Methods, Figure 2). Therefore, these cells appear to have progressed from their cell cycle arrest to meiotic prophase without DNA replication.”

*2) The Introduction needs some rewriting. For example, readers would benefit from a more thorough review of the* C. elegans *literature and that of* Drosophila*, as pertains to how nutrition affects germ cells.*

Good suggestion. We have expanded the Introduction to include a more thorough review of how nutrition affects germ cells in larval *C. elegans* and in *Drosophila*.

*3) In the Introduction, or in the Results section (near the end of the subsection “During starvation, germ cells progress slowly through S phase and arrest in G2”), point out that PGCs arrest in G2 during embryogenesis and remain arrested in G2 when the newly hatched larvae are starved, only undergoing proliferation upon feeding (Fukuyama et al., 2006).*

Another good suggestion. We have added this information to our expanded Introduction.

*4) Figure 7 and the subsection “Starvation-induced quiescence is distinct from cell-cycle arrest induced by DNA-damage”: that the "phosphor-inhibited CDK-1" antibody reports only P-CDK-1 and does not cross-react with unphosphorylated CDK-1 in the immunostaining experiment has not been validated here or in Craig et al. (2012). The authors should present a validation of this reagent.*

The phospho-specificity of our phospho-CDK-1 antibody was validated in *C. elegans* embryos by Rahman et al. PNAS 2014, which we cite in our Methods section. We have now additionally validated the phospho-specificity of this antibody in the *C. elegans* germline, by showing that staining disappears in germlines treated with lambda protein phosphatase (Figure 7—figure supplement 1).